# Proteogenomic Analysis Provides Novel Insight into Genome Annotation and Nitrogen Metabolism in *Nostoc* sp. PCC 7120

Shengchao Yu,[a,b] Mingkun Yang,[a,b] Jie Xiong,[a,b] Qi Zhang,[a,b,c] Xinxin Gao,[a,b] Wei Miao,[a,b] [iD] Feng Ge[a,b]

[a]State Key Laboratory of Freshwater Ecology and Biotechnology, Institute of Hydrobiology, Chinese Academy of Sciences, Wuhan, China
[b]University of Chinese Academy of Sciences, Beijing, China
[c]College of Fisheries and Life Science, Dalian Ocean University, Dalian, China

Shengchao Yu and Mingkun Yang contributed equally to this article. Author order was determined by their discussion.

**ABSTRACT** Cyanobacteria, capable of oxygenic photosynthesis, play a vital role in nitrogen and carbon cycles. *Nostoc* sp. PCC 7120 (*Nostoc* 7120) is a model cyanobacterium commonly used to study cell differentiation and nitrogen metabolism. Although its genome was released in 2002, a high-quality genome annotation remains unavailable for this model cyanobacterium. Therefore, in this study, we performed an in-depth proteogenomic analysis based on high-resolution mass spectrometry (MS) data to refine the genome annotation of *Nostoc* 7120. We unambiguously identified 5,519 predicted protein-coding genes and revealed 26 novel genes, 75 revised genes, and 27 different kinds of posttranslational modifications in *Nostoc* 7120. A subset of these novel proteins were further validated at both the mRNA and peptide levels. Functional analysis suggested that many newly annotated proteins may participate in nitrogen or cadmium/mercury metabolism in *Nostoc* 7120. Moreover, we constructed an updated *Nostoc* 7120 database based on our proteogenomic results and presented examples of how the updated database could be used to improve the annotation of proteomic data. Our study provides the most comprehensive annotation of the *Nostoc* 7120 genome thus far and will serve as a valuable resource for the study of nitrogen metabolism in *Nostoc* 7120.

**IMPORTANCE** Cyanobacteria are a large group of prokaryotes capable of oxygenic photosynthesis and play a vital role in nitrogen and carbon cycles on Earth. *Nostoc* 7120 is a commonly used model cyanobacterium for studying cell differentiation and nitrogen metabolism. In this study, we presented the first comprehensive draft map of the *Nostoc* 7120 proteome and a wide range of posttranslational modifications. In addition, we constructed an updated database of *Nostoc* 7120 based on our proteogenomic results and presented examples of how the updated database could be used for system-level studies of *Nostoc* 7120. Our study provides the most comprehensive annotation of *Nostoc* 7120 genome and a valuable resource for the study of nitrogen metabolism in this model cyanobacterium.

**KEYWORDS** cyanobacteria, genome annotation, nitrogen metabolism, *Nostoc* sp. PCC 7120, proteogenomics

Address correspondence to Wei Miao, miaowei@ihb.ac.cn, or Feng Ge, gefeng@ihb.ac.cn.

Cyanobacteria, essential contributors to primary productivity and the evolution of life on the Earth, are ancient Gram-negative bacteria capable of performing oxygen-evolving photosynthesis (1, 2). They constitute a phylogenetic group with diverse forms and extensive ecological distribution. *Nostoc* sp. PCC 7120 (hereafter referred to as *Nostoc* 7120) is a typical filamentous photoautotrophic cyanobacterium that differentiates semiregularly spaced (every 10 to 15 cells) heterocysts along the filaments for

the fixation of atmospheric nitrogen when deprived of combined nitrogen (3). The carbon source provided by photosynthetic vegetative cells and the nitrogen source provided by nitrogen-fixing heterocysts enable the filaments to achieve nutritional balance and grow normally under nitrogen deficiency (4, 5). This unique metabolic mode and specialized cell type make *Nostoc* 7120 an ideal model organism to study biological nitrogen fixation, cell differentiation, and multicellular pattern, and it has potential as a bioreactor for metabolic engineering (6–9).

The complete genome of *Nostoc* 7120 was sequenced and assembled in 2001 with the 7.21-Mb sequences comprising a circular chromosome and six plasmids designated pCC7120$\alpha$, pCC7120$\beta$, pCC7120$\gamma$, pCC7120$\delta$, pCC7120$\varepsilon$, and pCC7120$\zeta$, with a total of 6,310 putative protein-coding genes (10). The completion of the whole-genome sequencing of this strain has greatly promoted the investigation of the differentiation mechanism of cyanobacteria from both the physiological and genetic aspects, especially the differentiation of heterocysts. However, after nearly 2 decades of research updates, roughly 57.5% of the proteins in the CyanoBase reference protein database of *Nostoc* 7120 were still annotated as unknown protein or hypothetical protein (hereafter collectively referred to as putative protein) (http://genome.microbedb.jp/cyanobase/GCA_000009705.1). Therefore, a comprehensive and accurate genome annotation is urgently needed for system-level studies of *Nostoc* 7120.

The traditional genome annotation methods are highly dependent on computational algorithms and homology searches, which pose challenges for the annotation of species-specific potential genes, small open reading frames (ORFs), or ORF boundary definitions (11, 12). Owing to the lack of a proper correction evaluation mechanism, there are considerable proportions of pseudogenes and annotation errors in prokaryotic genome annotations, especially the annotation errors at the N terminus of proteins, which complicate the study of the physiological mechanisms of the corresponding species (13, 14). To prove the existence of a putative coding gene, the most direct evidence is that the coded product (that is, the target protein) can be detected. With the development of mass spectrometry (MS), high-quality peptides from large-scale proteomics experiments provide direct and corroboratory evidence for potential coding genes (15–18). Thus, an emerging research approach for improving genome annotation, called proteogenomics, has been derived from the integrative analysis of proteomic and genomic data. Different from the routine proteomics analysis strategy, collective proteomic data were matched against customized protein sequence databases generated from genomic or transcriptomic information to confirm the expression of predicated genes and help refine gene models (19, 20). Proteogenomics has become a particularly effective method to refine genome annotation of various organisms from viruses (21–23) to prokaryotes (24–27) and eukaryotes (28–33).

In this study, we performed an integrated proteogenomic analysis using high-resolution MS data to improve the genome annotation of *Nostoc* 7120. This is the first time that we provided protein-level evidence for nearly 90% of the predicted protein-coding genes in the *Nostoc* 7120 genome and reliably identified 26 novel genes and 75 revised gene models. We also cataloged large-scale protein posttranslational modifications (PTMs), including novel modifications such as lysine lactylation and benzoylation in cyanobacterium, which are involved in various metabolic processes of *Nostoc* 7120. We also constructed an updated database of *Nostoc* 7120 based on our proteogenomic results and presented examples of how the updated database could be used for system-level studies of *Nostoc* 7120.

## RESULTS

**Proteogenomic strategy for *Nostoc* 7120 genome reannotation.** The proteogenomic strategy for *Nostoc* 7120 genome annotation is shown in Figure 1. To obtain the condition-specific expression of proteins and diverse PTM types, we grew *Nostoc* 7120 under different cultural conditions. To achieve an in-depth coverage of the peptides and proteins, we processed samples using two different proteolytic digestion methods:

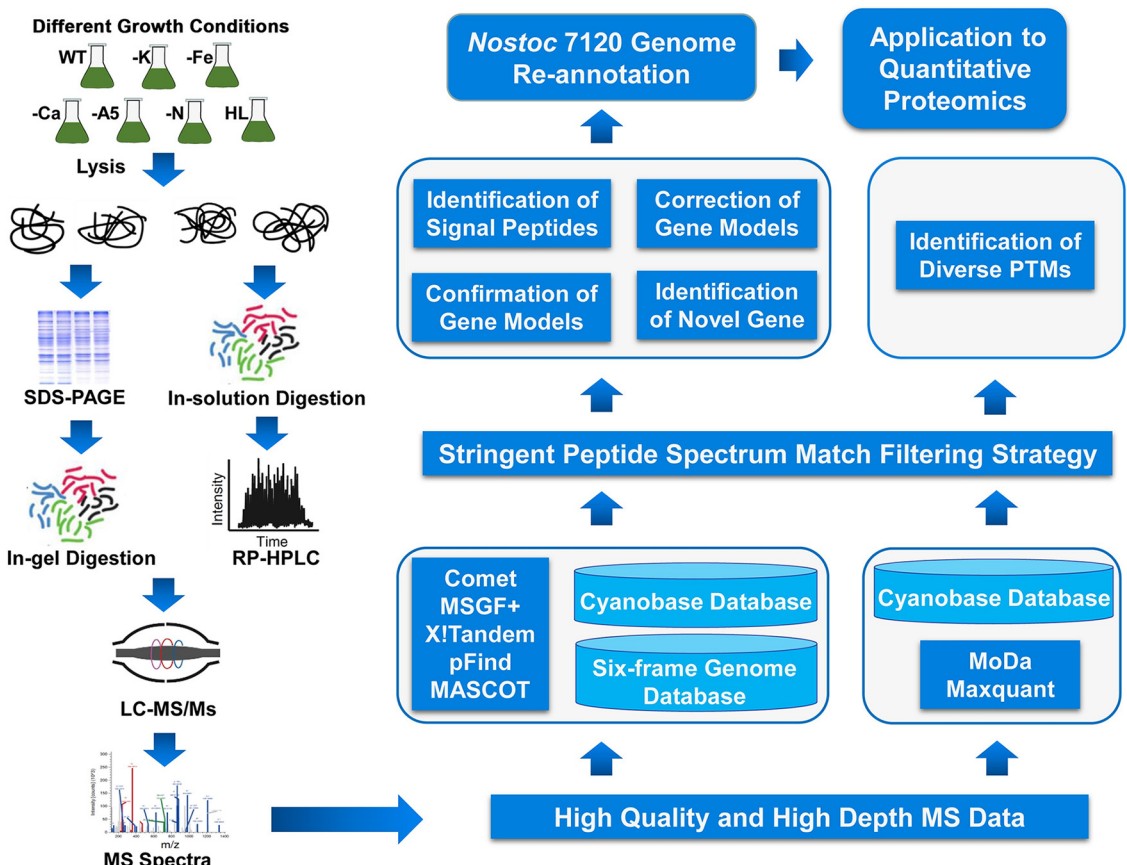

**FIG 1** Schematic workflow of proteogenomic analysis of *Nostoc* 7120. Protein extracts were prepared from *Nostoc* 7120 cultures under different conditions, including potassium deficiency (-K), iron deficiency (-Fe), calcium deficiency (-Ca), A5 deficiency (-A5), nitrogen deficiency (-N), and high light (HL) stresses. The proteins were digested and fractionated in gel and in solution, and the subsequent peptides were analyzed on high-resolution mass spectrometers. High-resolution mass spectrometry data were searched against a customized protein database using multiple search engines to refine the genome annotation of *Nostoc* 7120.

in-gel digestion and in-solution digestion. After fractionation, the generated peptides were subjected to MS analysis. Finally, all raw MS data were analyzed with the GAPP tool against the *Nostoc* 7120 proteogenomics databases (34). We also searched the MS data using two additional search algorithms (pFind [35] and MASCOT [36]) as separate steps, and the results were added into the GAPP pipeline to obtain a deep coverage. All raw MS data and identified peptides from different search engines are available at the iprox database (https://iprox.org) with the identifier IPX0002995000.

**Proteomic landscape of *Nostoc* 7120.** By applying a stringent false-discovery rate (FDR) threshold (FDR ≤ 1%), we identified a total of 97,738 unique peptides from five search engines. Among these, 94,264 peptides were mapped to the reference proteome database, resulting in the confirmation of translation of 5,519 (90%) previously predicted protein-coding genes (Table S1A and B). Our proteogenomic strategy unambiguously identified 5,519 proteins by at least two unique peptides or by a single peptide combined with manual validation, which confirmed 5,519 proteins already annotated in CyanoBase database. To our knowledge, this is the most comprehensive annotation of *Nostoc* 7120 genome to date (37, 38). Figure 2A depicts the distributions of all known peptides identified from each search engine. The number of peptides obtained from different search engines ranged from 49,744 to 68,905, with only 39,821 overlapping peptides. This observation suggested that the identified peptides could be greatly increased by integrating results from different search algorithms, in accordance with previous reports (39, 40). The distribution of the identified protein-coding sequences in the

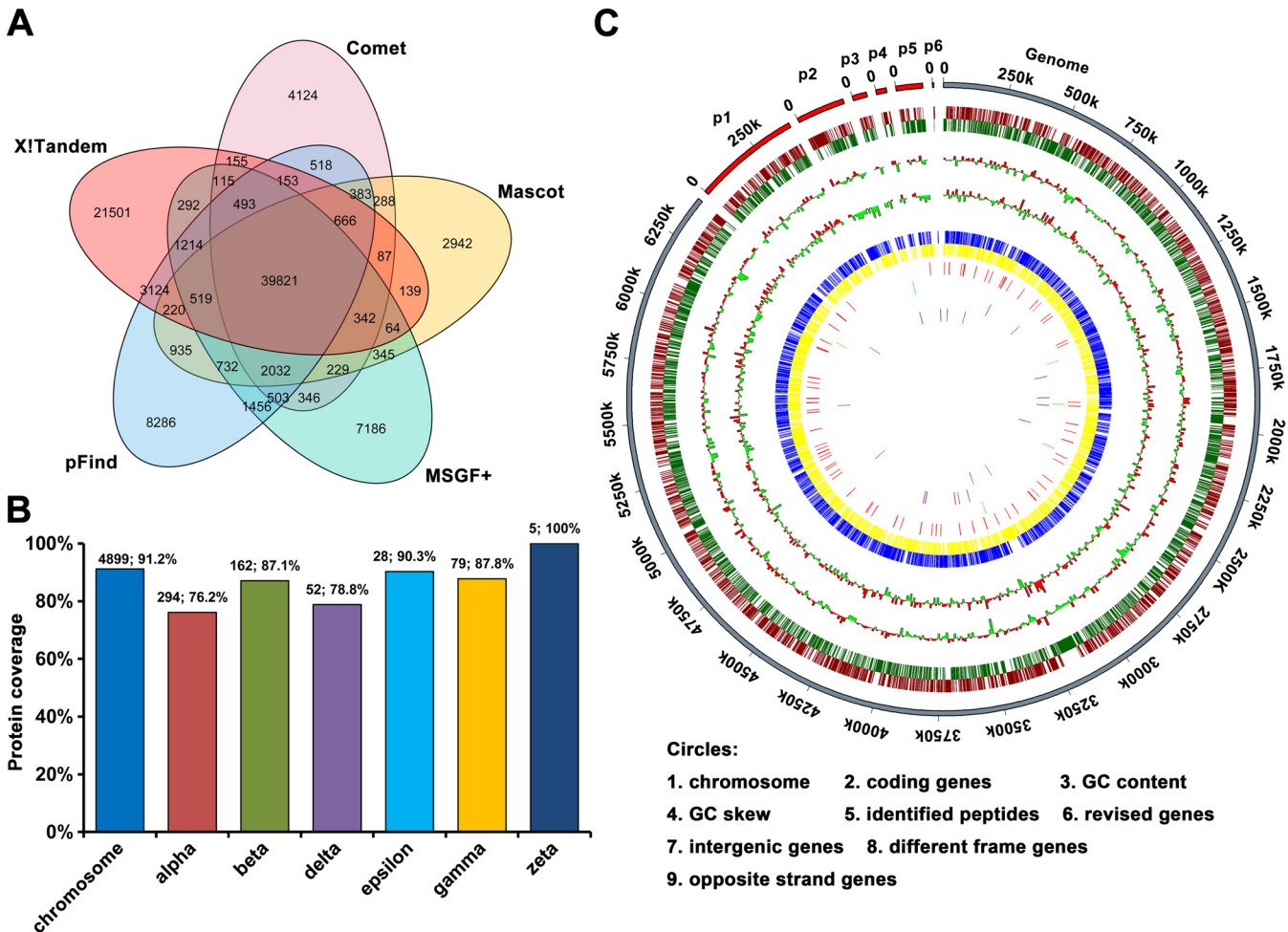

**FIG 2** Landscape of *Nostoc* 7120 proteomic identification. (A) Venn diagram showing the number of peptides identified using different search engines from the predicted proteins in CyanoBase. (B) Distribution of all identified proteins in the *Nostoc* 7120 chromosome and plasmids. (C) A circular plot showing the distribution of the identified peptides and proteins. The plot was created using Circos software.

*Nostoc* 7120 genome and different plasmids is shown in Figure 2B. Notably, over 90% of the identified proteins had two or more unique peptides, with an average coverage of 17 unique peptides per identified protein (Fig. S1A). The average sequence coverage per identified protein was 45.9%, and 41.9% of the identified proteins had peptide-level evidence for more than 50% of their sequences (Fig. S1B). These results confirmed the high quality of our MS data. To better visualize the proteomic results, we created a Circos plot to illustrate the global distribution of identifications in this study, including coding genes in the current CyanoBase, GC content, GC skew, all identified peptides, and proteins mapped to the chromosome and different plasmids (Fig. 2C).

**Identification and functional annotation of putative proteins.** As a large number (3,525 of 6,135) of the previously predicted protein-coding genes are annotated as putative proteins in the *Nostoc* 7120 genome, it seems that these proteins showed similarity only to hypothetical proteins or even showed no significant similarity to any known proteins (10). The proteome data can provide experimental evidence of translational expression of those putative genes at the protein level. In this study, we successfully identified 3,087 putative proteins, of which 381 were identified with one unique peptide and the others were identified with two or more unique peptides (Table S1C). Thus, our data suggested that at least 88% of the putative protein-coding genes in *Nostoc* 7120 genome had translational expression evidence.

To further systematically clarify the biological function of these putative proteins, we performed functional annotation for these proteins. As shown in Figure 3A, 67% of

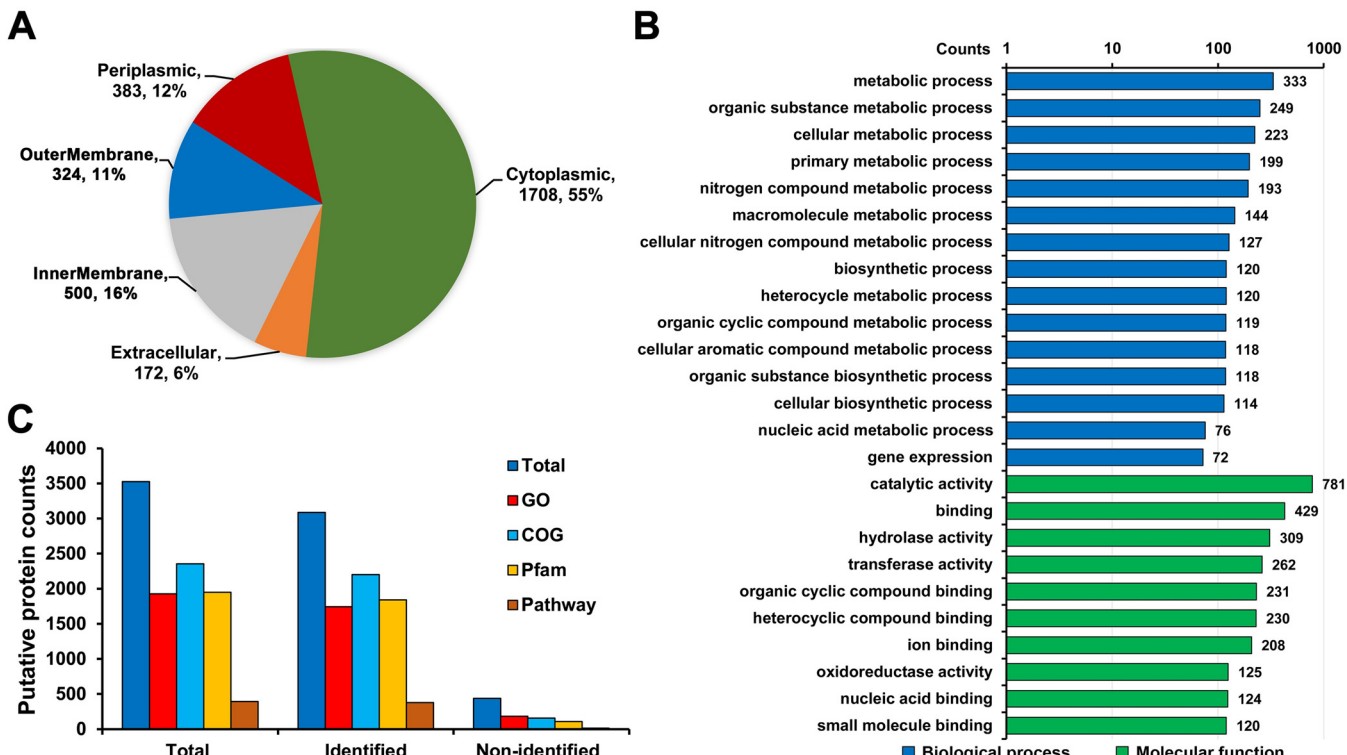

**FIG 3** Summary of the putative proteins identified in *Nostoc* 7120. (A) Pie chart showing the subcellular localization of identified putative proteins. (B) Bar graph depicting the Gene Ontology (GO) classification of all identified putative proteins, according to their biological processes and molecular functions. (C) Distribution of the functional annotation of total, identified, and nonidentified putative proteins.

the proteins were predicted to localize to the cytoplasm and periplasmic space, followed by the membrane (27%) and the extracellular space (6%), based on the subcellular localization analysis using the CELLO tool (Table S1D). Gene ontology (GO) classification showed that these putative proteins were involved in various metabolic pathways in the biological process category, such as biosynthetic and nitrogen metabolism, as well as enzymatic activity and binding to various targets within the molecular function category (Fig. 3B and Table S1E). Furthermore, the functional features of the identified putative proteins were analyzed based on GO, Clusters of Orthologous Genes (COG), KEGG pathway, and Pfam annotations, and we noticed that 2,450 putative proteins had one or more piece of functional evidence (Fig. 3C and Table S1E). These results will provide a valuable resource for future efforts to explore the function of these proteins in the metabolic networks of *Nostoc* 7120.

**Characteristics of unidentified proteins.** Despite the in-depth coverage of our MS data, 616 proteins were still not detected using any peptide evidence in our data. Among these, 118 proteins (shared proteins) matched only a subset of peptides shared by two or more proteins, which were not included in the identified set because there was no unique peptide evidence (Table S2A). We next analyzed these redundant proteins based on amino acid sequence identity. It was important to note that 81 of 118 shared proteins were redundant, corresponding to 20 proteins, although they were assigned to different accession numbers from the CyanoBase database (Table S2B). According to the annotation of these shared proteins, we found that 68 were annotated as transposases. In addition, there were three groups of heterocyst-associated proteins, originally encoded by *fdxN*, *nifD*, and *hupL* genes, which have been annotated with a C-end or N-end label to indicate the genome rearrangement in *Nostoc* (10) (Table S2B). Of the 616 unidentified proteins, 438 were annotated as unknown or hypothetical proteins in the CyanoBase database (Table S2C). Functional annotation revealed that 232 putative proteins (approximately 54%) had functional features, according to their GO, COG, KEGG pathway, and Pfam annotations

(Table S2C). Considering the challenge for the detection of microproteins (≤100 amino acids) in traditional MS approaches, we hypothesized that the length of the microproteins could influence protein identification owing to the low abundance of peptides generated from these proteins (41). We found that 54% (335/616) of the unidentified proteins were short (<100 amino acids) and 9 were shared proteins (Fig. S1C and Table S2D). Together, the union set of unidentified proteins caused by shared peptides, putative protein categories, or low molecular weight contained 573 proteins, comprising 93.0% of the total unidentified proteins.

**Identification of global posttranslation modifications in *Nostoc* 7120.** Chemical modification at specific amino acids of proteins is one of the most important PTMs and usually has diverse functions (42). However, little is known about the PTMs in *Nostoc* 7120. In this study, we identified proteome-wide PTMs based on our MS data; we identified a total of 8,200 modification sites corresponding to 27 distinct PTM events from 2,297 proteins (Fig. 4A and Table S3) by using a previous reported search strategy (28, 43). Among these identified PTMs, the three most abundant PTMs were methylation (K/R/E/Q/C), glutarylation (K), and propionylation (K), corresponding to 3,039 methylation sites, 822 glutarylation sites, and 575 propionylation sites, respectively (Fig. 4A and Table S3). Notably, the deep-coverage MS data enabled us to discover three acylation modifications, including benzoylation (K), lactylation (K), and glutarylation (K), in cyanobacteria for the first time.

To further investigate the function of our discovered modified proteins, we next performed functional enrichment analysis of GO and KEGG pathway. As indicated in Figure 4B, these modified proteins were predominantly enriched in photosynthesis-related components and involved in various metabolic processes, including nitrogen compound metabolism (Table S4A and B). We also further illustrated the potential metabolic regulation of these PTM events by mapping the modified proteins to KEGG pathways. Consistent with the functional annotations, we noticed that most modified proteins were involved mainly in metabolic pathways, followed by ABC transporters, ribosome, photosynthesis, and two-component system (Table S4C), suggesting that these PTMs may play regulatory roles in metabolism, transport, protein translation, and stress response. Figure S2 exhibits the modified proteins involved in diverse carbon metabolism and nitrogen metabolism pathways in *Nostoc* 7120. For example, many photosynthesis-related proteins and a series of key enzymes participating in carbon and nitrogen metabolism were modified with different types of PTMs. In addition, we found that the key regulators of heterocyst development, such as CcbP, NtcA, and HetN, have also undergone diverse PTMs. We also performed Western blotting to experimentally verify some of the PTMs identified in this study. As expected, strong immunoblot signals were detected, indicating that these modifications were abundant in *Nostoc* 7120. It was worth noting that some acylation modifications closely related to metabolic regulation, such as acetylation, propionylation, crotonylation, etc., showed signal differences under normal or nitrogen deprived conditions, which further indicated that these modifications may be involved in the regulation of nitrogen metabolism in the Cyanobacteria (Fig. 4C). In summary, our data provide a rich source to understand the biological functions of these PTMs.

**Signal peptide prediction.** Based on the subcellular localization analysis of the identified 5,519 previously predicted protein-coding genes in *Nostoc* 7120, roughly 42% were predicted to be located in the periplasm, in the membrane, and outside the cell (Table S1D). In addition, the functional annotation of modified proteins revealed that many proteins were involved in ABC transport process and predicted to localize to the membrane. Consequently, in addition to posttranslational chemical modifications, we hypothesize that a signal peptide that is important for protein maturation, subcellular localization, and secretion may exist (44). To test this hypothesis, two signal peptide prediction tools, PrediSi (45) and SignalP (46), were used to analyze the signal peptides of all identified proteins. As illustrated in Figure S3A, there were 1,013 signal peptides predicted by PrediSi and 550 signal peptides predicted by SignalP, of which 414 putative signal peptides were supported by both prediction models (Table S5). As

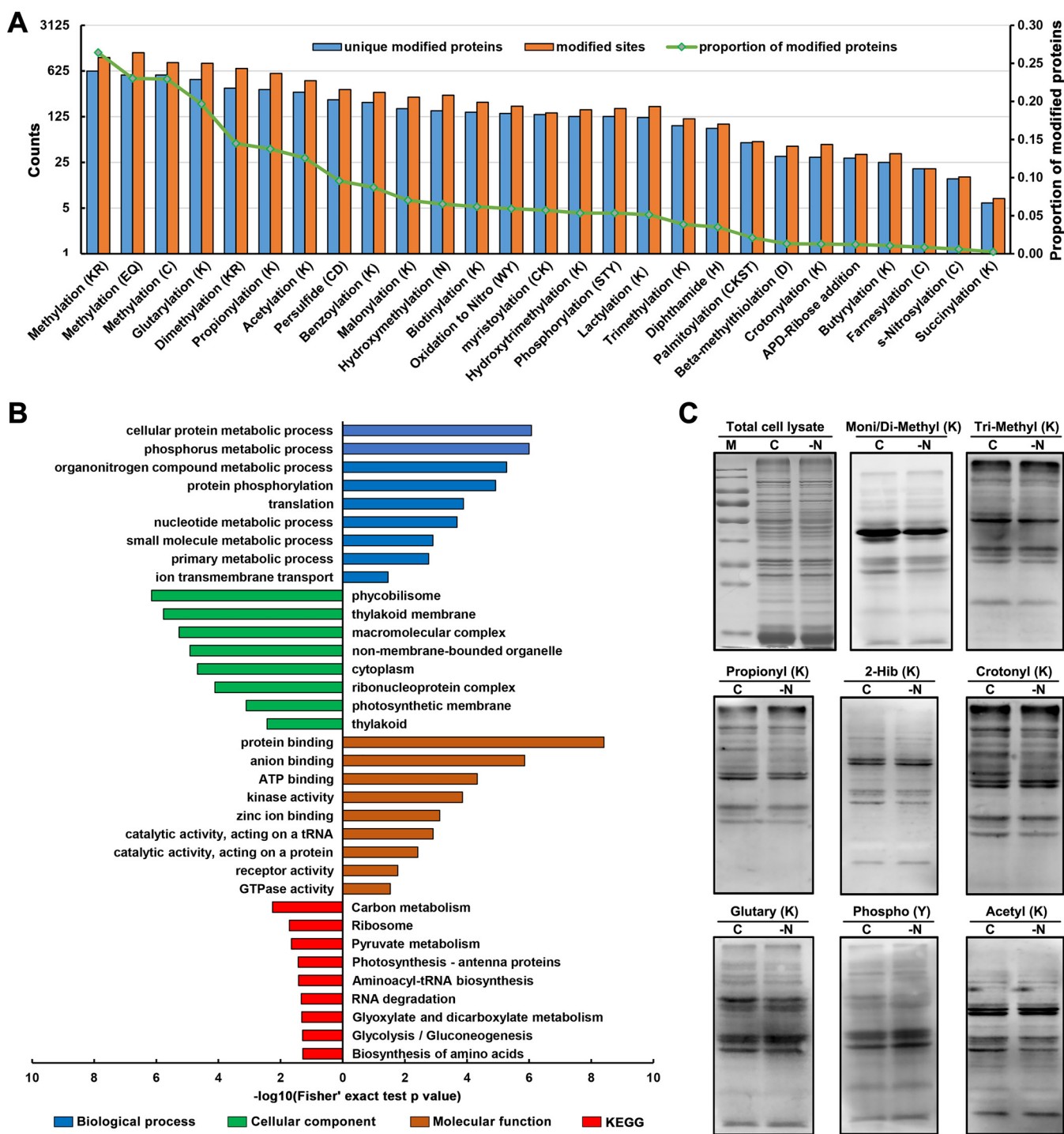

**FIG 4** Summary of protein PTM events in *Nostoc* 7120. (A) Histogram representation of the distribution of the number of proteins and sites from 27 types of PTMs discovered in *Nostoc* 7120. (B) Functional enrichment bar graph of GO and KEGG pathway for modified proteins. (C) Proteome-wide PTM levels in *Nostoc* 7120 after nitrogen deprivation compared with normal conditions. C, normal condition; -N, nitrogen deprivation; mono/di-methyl (K), mono/dimethylation (lysine); tri-methyl (K), trimethylation (lysine); propionyl K, propionylation (lysine); 2-Hib (K), 2-hydroxyisobutyrylation (lysine); crotonyl K, crotonylation (lysine); glutaryl K, glutarylation (lysine); phospho Y, phosphorylation (tyrosine); acetyl K, acetylation (lysine).

signal peptides are cleaved and quickly degraded to produce mature proteins, the signal peptide sequence may be difficult to detect using conventional proteomic approaches. Thus, the predicted signal peptide sequences that included the peptides identified from MS results would be refuted. We utilized our MS data sets to experimentally confirm the signal peptides that were predicted using *in silico* analysis and refuted 144 predicted signal

peptides by SignalP, as well as 338 predicted signal peptides by PrediSi (Fig. S3B). Further subcellular localization analysis revealed that these validated proteins containing signal peptides were located mainly in the periplasm, membrane, and extracellular space (Fig. S3C and D). Moreover, as expected, these signal peptide sequences showed the clear sequence motif with conserved alanine and valine residues at the −1 and −3 positions, similar to that observed in other prokaryotes (47) (Fig. S3E and F).

**Discovery of novel genes and gene model revisions in *Nostoc* 7120 genome.** Apart from the identification of previously predicted protein-coding genes, the peptides exclusively matching the unique locations in the *Nostoc* 7120 genome were designated genome search-specific peptides (GSSPs) and used for annotated gene model correction or novel gene discovery. In total, 3,474 unique orphan peptides were identified through a stringent filtering threshold (FDR ≤ 1%). Details of the credible GSSPs identified using five different search engines are listed in Table S6A, and the distribution of these GSSPs obtained from different search engines is shown in Figure S4. Finally, the GSSPs were mapped exclusively to the unique genomic locus to identify open reading frames (ORFs) using BLAST. ORFs that were mapped only to the genome regions without overlap with any known gene models were designated novel protein-coding genes. A total of 26 novel protein-coding genes, that is, the newly identified proteins, were identified with at least two unique GSSPs. Among the 26 novel potential protein-coding genes, 9 were located in the intergenic region, 6 were mapped to the different reading frame regions, and the remaining genes were mapped to the opposite strand of the existing genes (Table S6B). ORFs that partially overlapped the annotated protein-coding regions were revised genes, and we corrected 75 existing gene models, which were also supported by at least two unique GSSPs (Table S6C). Circles 6 to 9 in the circular proteome map in Figure 2C represent the distribution of all novel proteins and revised protein models, along with their pieces of supporting peptide evidence identified in this study.

Figure 5A depicts an example of a novel intergenic gene (NG-8), represented by three unique peptides that were mapped to the previously unannotated genomic region between *asr9501* and *asr9503* on the plasmid pCC7120zeta. This novel intergenic gene showed 100% sequence identity with a hypothetical protein, DSM107007_57930 (RUR72243.1), annotated in 2019 genome assembly of *Nostoc* 7120 by Will et al. (48), with additional RNA-seq evidence further supporting the identification of the intergenic gene in this study. Notably, this hypothetical protein, DSM107007_57930, was missing in the current genome annotation of CyanoBase. This observation further supported the discovery of our proteogenomic analysis. In addition, we detected a novel frame-shifted gene (NG-15) that was partially mapped to an annotated gene (*alr8555*) on the plasmid pCC7120delta (Fig. 5B). This gene was annotated in the 2019 and 2020 genome annotations of *Nostoc* 7120; it also showed high sequence identity with the protein annotated in the other species and was supported by transcriptomic evidence. Figure 5C shows the discovery of a novel gene (NG-20) carried by the opposite strand region of an existing gene. We mapped seven unique GSSPs to the opposite strand that was previously the unannotated genomic region of *asr3266* gene, with corroborating transcriptomic data being shown. Notably, protein sequence alignment analysis revealed that this protein shared 100% sequence identity with an existing protein annotated in the 2020 genome of *Nostoc* 7120, which was also missed in the current genome annotation of CyanoBase. In addition, the orthologous proteins of NG-20 were discovered in other organisms by using protein BLAST online. Finally, sequence alignment and functional analysis revealed that eight of these novel genes had been annotated in *Nostoc* 7120 genome annotation of the 2019 version or 2020 version and five novel genes had GO annotation (Table S6D). The in-depth MS data also enabled us to refine the boundary of predicted protein-coding gene in the current genome annotation of *Nostoc* 7120. In this study, we corrected 75 gene models of N-terminal extension (Table S6C). Figure 5D depicts a representative correction of a gene structure by extension of the N terminus. Eight unique peptides were mapped to the upstream region of the existing gene *alr5269*, and one

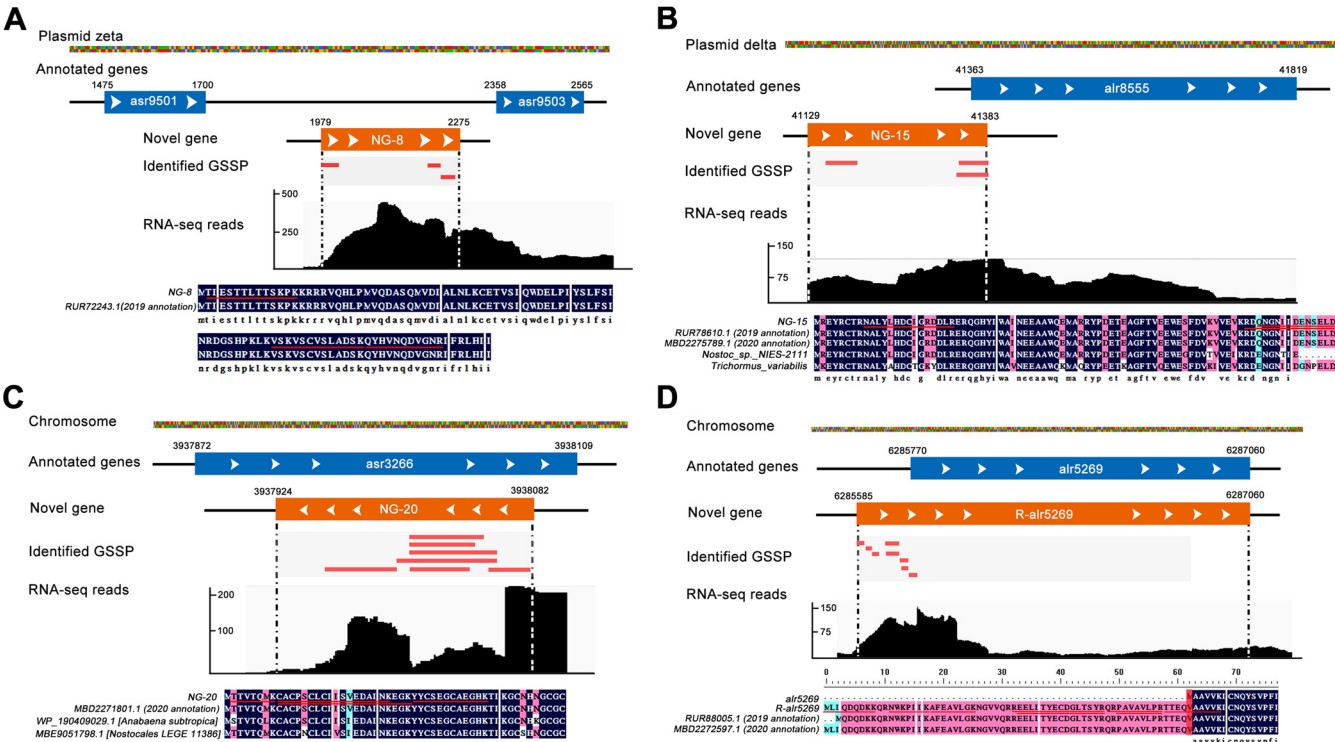

**FIG 5** Identification of novel genes and revised gene models. (A) Identification of an intergenic novel gene (NG-8). Three novel peptides mapped to the intergenic region in the *Nostoc* 7120 genome. (B) Detection of a novel gene (NG-15) based on three novel peptides mapping to a different reading frame region of alr8555 and RNA-seq evidence. (C) Identification of a novel gene (NG-20) based on seven novel peptides mapping to an opposite strand of the existing gene-annotation region of asr3266 and RNA-seq evidence. (D) Revision of the gene model. Eight novel peptides partially mapped to the upstream region of an annotated gene (alr5269). 2019 annotation and 2020 annotation represented that the sequence existed in the *Nostoc* 7120 genome annotation of the 2019 version or the 2020 version in NCBI database, respectively.

GSSP spanned the initiation site of the N terminus of *alr5269*. It is worth noting that the correction model of *alr5269* was also reported by other groups according to the NCBI-Ref-seq database of *Nostoc* 7120, further implying the high quality of our MS data. Moreover, the obtained correction of the existing gene was supported by transcriptomic evidence, and we observed homologous genes for this annotation in other species (Fig. 5D). GO annotations revealed that these revised genes were distributed in a wide range of functional classes (Table S6E). Additional sequence alignment analysis indicated that there were still differences in many gene boundaries, although some revised genes shared 100% sequence similarities with the proteins annotated in the 2019 version or 2020 version of the genome (Table S6E). Furthermore, we discovered several noncanonical translation initiators among the identified 101 novel events. For instance, we identified eight revised genes that used ATA and ATT as the start codon and two novel genes that used ATA as the start codon (Table S6).

**Validation of novel genes.** To further validate the novel findings in this study, we independently validated our results using published RNA-seq data (49), real-time PCR (RT-PCR) analysis, and MS analysis of synthetic peptides. First, using transcriptome analysis, 25 novel genes, excluding the NG-19 gene, were identified as differentially expressed under nitrogen starvation conditions. We constructed a heatmap showing the relative levels of each gene based on Z-score (Fig. 6A). In addition, the expression of these novel genes under nitrogen stress conditions was further verified by RT-PCR (Fig. 6B). We found that most novel genes were upregulated after nitrogen deprivation for 3 and 12 h and significantly downregulated at 24 h. Although there was divergence between transcriptomic and RT-PCR due mainly to the different samples used, these results not only showed the differential expression of the identified novel genes in this study but also provided the transcript-level evidence for the existence of these novel genes, suggesting that they may be involved in the response to nitrogen stress in *Nostoc* 7120.

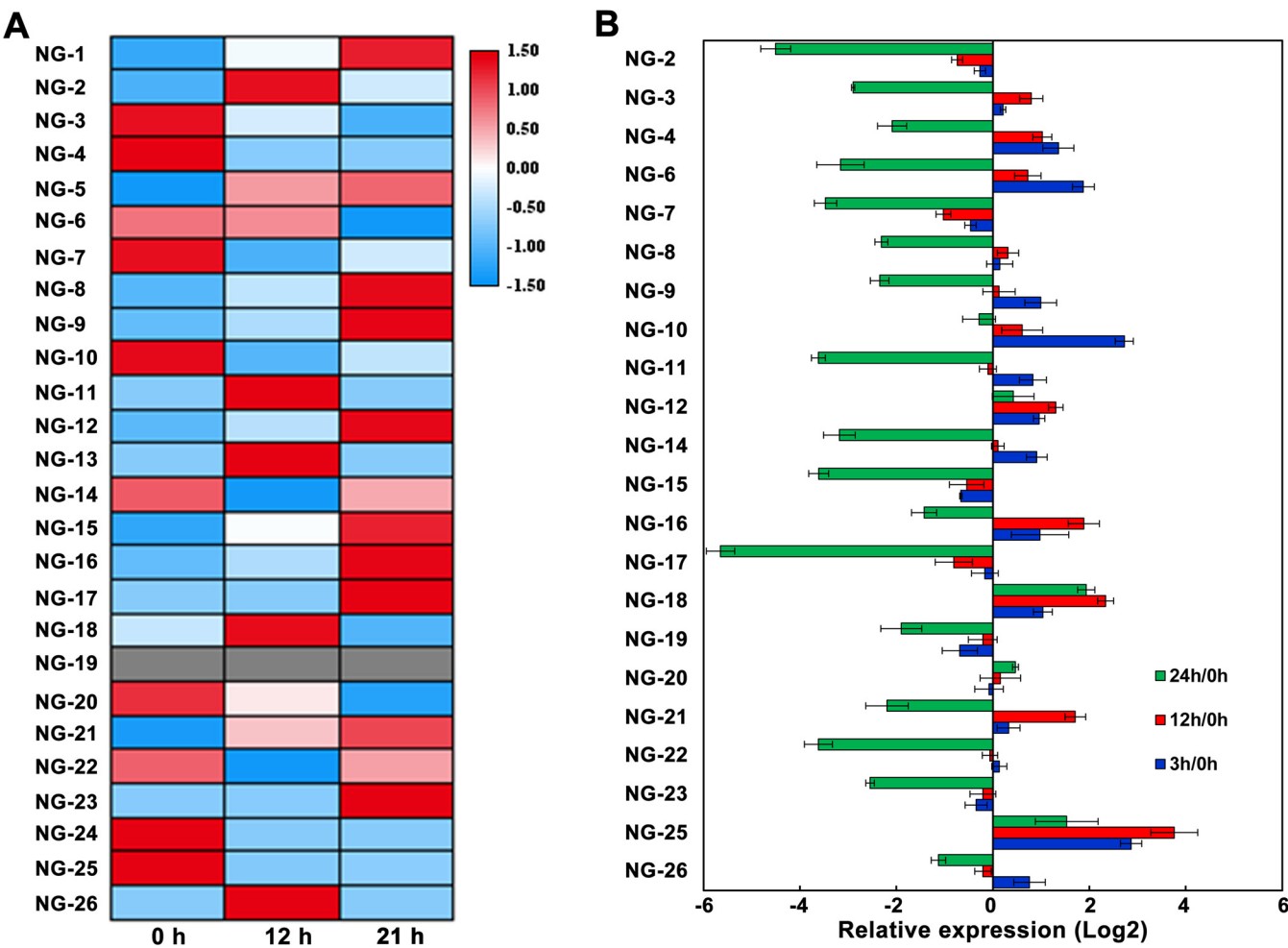

**FIG 6** Validation of novel genes. (A) Heatmap showing the relative mRNA expression levels of 26 novel genes in response to nitrogen deficiency, according to RNA-seq data. (B) RT-PCR validation of novel genes. RNAs were extracted from *Nostoc* 7120 grown under nitrogen deficiency.

To experimentally validate the novel genes further, we synthesized 25 random GSSPs and analyzed them via MS (Table S7A). By comparison of the results of the mass spectra for these peptides, we could confirm the reliability of a subset of the identifications in our proteogenomic analysis. We found that the relative intensity and distribution of b/y ions in the synthesized peptide spectra was highly consistent with the corresponding GSSPs identified from proteogenomic analysis, strongly supporting our proteogenomic data. The representative tandem mass spectrometry (MS/MS) spectra for the four synthesized peptides and GSSPs are shown in Figure S5. The remaining spectra are available at https://iprox.cn.org with the identifier IPX0002995000 (under the file name "MS validation of GSSPs by synthesized peptides").

**Quantitative proteomics analysis of *Nostoc* 7120 under nitrogen starvation conditions.** Considering the discovery of 101 reliable novel events (26 novel proteins and 75 revised protein models) in *Nostoc* 7120, we determined the expression levels and provided additional evidence for these novel events using tandem mass tag (TMT)-based quantitative proteomics analysis. The TMT-based quantitative proteomic data sets were searched against the current CyanoBase database and the updated database, respectively. As shown in Figure 7A, 3,315 protein groups were identified and quantifiable in our updated database owing to the presence of novel genes. After filtering with unique peptide (≥2), we identified more proteins with unique peptides ≥2 in the updated database (2,835) than in the current CyanoBase (2,814). Furthermore, among the identified proteins, 50% (1,339) were annotated as putative proteins in the current CyanoBase, whereas the identified 2,569 proteins had functional features in our

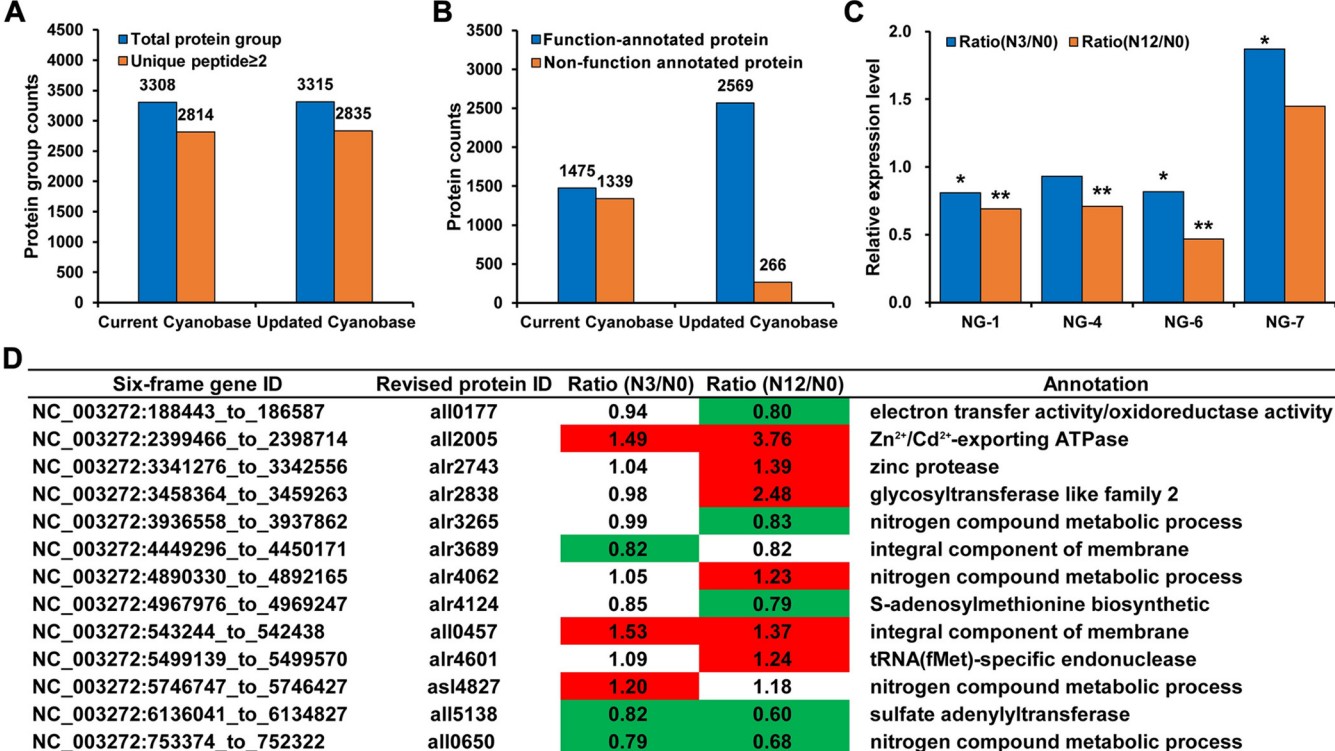

**FIG 7** TMT-based quantitative proteomic analysis using the current CyanoBase and the updated CyanoBase. (A) Histogram showing the number of proteins identified in the current CyanoBase and the updated CyanoBase. (B) Functional annotation of proteins identified in the current CyanoBase and the updated CyanoBase. (C) Relative expression levels of four novel genes under nitrogen deficiency. *, $P < 0.05$; **, $P < 0.01$. (D) Relative expression levels of the revised genes under nitrogen deficiency.

updated database, according to their function annotation, especially for putative proteins as described in "Identification and functional annotation of putative proteins" above (Fig. 7B). Only 266 quantifiable proteins were still annotated as putative proteins (Fig. 7B). Moreover, based on the thresholds of fold change of $\geq 1.2$ and $P$ value of $\leq 0.05$, three novel proteins, NG-1, NG-4, and NG-6, were found to be significantly downregulated, while NG-7 was upregulated under nitrogen stress (Fig. 7C). Notably, the protein level measured by quantitative proteomics analysis had a poor correlation with the corresponding RNA level detected by transcriptome or RT-PCR, which may be mediated by a series of regulatory processes, including posttranscription, translation, and protein degradation regulation. In addition to the identified novel proteins, 14 revised proteins were also differentially expressed under nitrogen stress and involved in different biological processes, such as nitrogen compound metabolic process (Fig. 7D). Consequently, our quantitative proteomic data could be used to estimate differences in protein abundance under nitrogen stress and provide additional expression evidence for novel events in this study. Details of the quantitation results are presented in Table S8. The MS proteomics data have been deposited at the iprox database (https://iprox.org) with the identifier IPX0002995002.

## DISCUSSION

In this study, we provided robust protein-level evidence for approximately 90% (5,519) of the previously predicted protein-coding genes in *Nostoc* 7120 using a comprehensive proteogenomic analysis. Among them, 2,450 putative proteins were evidenced by the presence of a subset of peptides identified in our MS data and had functional features in our updated database based on our bioinformatic analysis. Moreover, our comprehensive strategy enabled us to unambiguously identify a series of novel events, including 26 novel genes and 75 revised gene models, as well as a diverse set

of PTMs. Our study offered the most comprehensive annotation of the *Nostoc* 7120 genome.

In this study, 616 previously predicted protein-coding genes were missed in our detection data set. Among them, 118 were shared proteins, which were excluded in the identified set owing to the absence of unique peptide evidence. In the unidentified protein set, we found 326 microproteins ($\leq$100 amino acids). Furthermore, the identification percentage of 553 microproteins in our MS data (62%) was much lower than that of large proteins (length of over 100 amino acids) (Fig. S1C). It seems that an appropriate identification strategy should be applied for the analysis of microproteins, such as enrichment-based small proteins detection (50, 51) and digestion with different proteases (52), to generate a more comprehensive coverage of the *Nostoc* 7120 proteome.

The presence of prediction bias for small ORFs in automated genome annotation tools may be one of the possible reasons why a subset of small ORFs were overlooked in previous *in silico* predictions (53, 54). Apart from this, small proteins are often difficult to experimentally verify, indicating that they are often overlooked in genome annotation. It has been reported that small ORFs are a group of functionally important proteins that may participate in various biological processes, such as transcriptional regulation (55, 56), development (57), signal transduction (55), and metabolism (58), in a wide range of organisms. For example, in *Nostoc* 7120, the microprotein/micropeptide PatS was reported to be involved in regulating the formation and development of heterocysts (59). Therefore, considering the functional importance of these proteins, we analyzed the identified proteins using our proteogenomic analysis, and the results revealed that the sequence length of 24 novel proteins was less than 100 amino acids. Additional functional annotation and quantitative proteomics analysis revealed that these proteins may play roles in the response to stress conditions, suggesting the potential function of these small ORFs in *Nostoc* 7120. Although further studies are required to assess the functional relevance of these novel proteins, these small ORFs were differentially expressed under nitrogen starvation conditions, indicating that these proteins may participate in the regulation of nitrogen metabolism or heterocyst formation of *Nostoc* 7120.

The determination of the translation initiation site of a gene or N terminus of a protein is an important step in genome annotation. Owing to the limitation of gene-prediction algorithm and presence of noncanonical translation initiators, translation initiation site confirmation is still a challenge, particularly for prokaryotes (13, 60). Importantly, a previous study demonstrated that approximately 8% to 9% of proteins in the cyanobacterial database contain an incorrect N terminus (61). Therefore, in addition to identifying missed small ORFs in this study, we also revised the annotated gene models in the current CyanoBase, and a total of 75 N termini of the annotated proteins were experimentally corrected with the peptide evidence obtained from our proteogenomic data. Among the revised initiation codons, in addition to the typical initiation codons ATG, GTG, CTG, and TTG, noncanonical codons were also reported to serve as translation initiation codons, including ATT and ATA (61, 62). We identified eight revised genes with noncanonical translation initiation codons (ATA and ATT) (Table S6). These findings will improve the criteria and algorithms of computational predictions for genome annotation and facilitate genetic manipulation, as well as comprehension of the structural and functional characteristics of these revised genes in *Nostoc* 7120.

In the current version of CyanoBase, approximately 60% of the *Nostoc* 7120 proteins are annotated as hypothetical or unknown proteins. In this study, we annotated 86% of proteins with specific biological functions using a bioinformatic strategy. For example, six hypothetical proteins were annotated as nitrogenases through domain analysis, based on the Pfam database (Table S1E), which will contribute to the further investigation of the structural composition of oxygen-sensitive nitrogenase complex and nitrogen metabolism in heterocysts (63). Moreover, a large proportion of hypothetical or unknown proteins were assigned to tetratricopeptide repeat proteins (42 putative proteins) and pentapeptide repeat proteins (33 putative proteins) (Table S1E), which were associated with the biogenesis of the photosynthetic apparatus (64) and heterocyst

differentiation (65) in *Nostoc* 7120. Based on our analysis, we constructed an updated CyanoBase including novelties, functional annotations, PTMs, and signal peptides. We expect that this updated database will be a valuable resource for system-level studies in *Nostoc* 7120.

Cadmium (Cd) and mercury (Hg) are two common toxic heavy metals in contaminated aquatic environments, and water pollution by Cd/Hg ions is an environmental problem worldwide (66). In this study, we identified a novel protein, NG-20, and annotated it as a metallothionein based on sequence alignment analysis (Fig. 5C). Consistent with our results, NG-20 has been reported to play a key role in Cd tolerance in *Nostoc* 7120 (67). Additionally, a number of Cd/Hg metabolism-related proteins were also identified using protein sequence alignment and functional annotation (Table S9). Among them, 10 previously annotated as putative proteins identified in this study were involved in the efflux system of Cd ions, indicating that these proteins may participate in managing the intracellular Cd ions in *Nostoc* 7120. Based on our functional analysis and previous reports (68), we summarized the role of Cd/Hg-related proteins in different pathways in *Nostoc* 7120 (Fig. S6). The identification of Cd/Hg-related proteins in *Nostoc* 7120 could provide the basis for the development of a synthetic biological approach to enhance Cd/Hg bioremediation.

In conclusion, the present study represents the most comprehensive annotation of the *Nostoc* 7120 genome to date. The identified novel events and holistic protein PTMs provide a valuable resource for further mechanistic investigation of nitrogen or Cd/Hg metabolism in this model cyanobacterium.

## MATERIALS AND METHODS

**Cell culture and protein extraction.** The wild type of *Nostoc* 7120 was cultivated under several conditions and harvested at different growth stages and treatments. Briefly, the wild-type strain was cultured in liquid medium BG11 bubbling with filtrated air at 30°C under continuous illumination at 30 $\mu$mol photons m$^{-2}$ s$^{-1}$. The cells were grown to log phase (optical density at 730 nm [OD$_{730}$] of ~0.8) and stationary phase (OD$_{730}$ of ~2.0) in liquid medium BG11. For stress treatments, the exponentially growing cells were then collected by centrifuging at 5,000 rpm for 10 min, washed three times using the fresh BG11 liquid medium (lacking the specified nutrient), and immediately resuspended in BG11 medium lacking the specified nutrient, including in potassium-free medium that contained Na$_2$HPO$_4$ instead of K$_2$HPO$_4$, in Fe-free medium that added Fe-binding chelator to the medium, in Ca-free medium that contained NaCl instead of CaCl$_2$, and in A$_5$-free medium [without the trace elements CuSO$_4$, H$_3$BO$_3$, ZnSO$_4$, MnCl$_2$, Co(NO$_3$)$_2$O, Na$_2$MoO$_4$]. Alternatively, the cells were exposed to the nitrogen deficiency medium BG11 without NaNO$_3$ and cultured for 0, 3, 12, and 24 h, respectively. For high-intensity light treatment, cells in the exponential phase were adjusted to an OD$_{730}$ of ~0.4 and then exposed to 400 $\mu$mol photons m$^{-2}$ s$^{-1}$ for 60 min. The cultures at different growths and treatments were harvested by centrifugation at 4,000 $\times$ *g* for 10 min at room temperature and washed twice with phosphate-buffered saline (PBS) buffer. Finally, cells were resuspended in lysis buffer (20 mM Tris-Cl [pH 7.5], 150 mM NaCl, 1% Triton X-100, 5 mM $\beta$-glycero-phosphate, 10 mM NaF, 1 mM Na$_3$VO$_4$, 10 mM Na$_4$P$_2$O$_7$, 50 mM nicotinamide [pH 7.5]) containing 1$\times$ protease inhibitor cocktail and 1$\times$ phosphatase inhibitor cocktail. The mixture was submitted to sonication (3 s on, 3 s off) for 30 min on ice with an output of 135 W (JY92-IIN; Ningbo Scientz Biotechnology Co., Ltd.). The remaining debris was removed by centrifugation at 6,000 $\times$ *g* at 4°C for 30 min. The supernatant was collected, and protein concentration was determined with bicinchoninic acid (BCA) kit (Beyotime Institute of Biotechnology, Jiangsu, China) according to the manufacturer's instructions.

**Trypsin digestion and prefractionation.** In-gel and in-solution proteolytic digestion were performed as described previously (28). Briefly, for the in-gel digestion, 1 mg protein extract was solubilized in 5$\times$ SDS loading buffer and denatured at 95°C for 5 min before fractionation by 12% SDS-PAGE gel. Gels were stained with Coomassie brilliant blue R250. Then gels were excised into 10 pieces, and each gel band was further cut into ~1-mm$^3$ fractions. The gel pieces were thoroughly destained with 35% acetonitrile and 50 mM NH$_4$HCO$_3$ and dehydrated with 75% (vol/vol) acetonitrile, followed by disulfide reduction with 25 mM dithiothreitol and alkylation with 50 mM iodoacetamide. Finally, proteins were digested with 50 mM NH$_4$HCO$_3$ containing 20 $\mu$g/ml sequencing grade modified trypsin (1:100, wt/wt) overnight at 37°C. The supernatants were collected and transferred into a new microcentrifuge tube. The remaining gels were sonicated twice with extraction buffer (67% [vol/vol] acetonitrile, 5% [vol/vol] trifluoroacetic acid). The peptide supernatants and extracts were mixed and desalted for further mass spectrometry analysis.

For the in-solution digestion, the protein extract was reduced with 25 mM dithiothreitol for 40 min at room temperature and alkylated with 50 mM iodoacetamide for 10 min at room temperature in darkness. Proteins were precipitated with 5 volumes of precooled acetone to remove pigment molecules or other small molecules as much as possible. The pellets were then redissolved with 50 mM NH$_4$HCO$_3$ and

digested with 1% trypsin (wt/wt) overnight at 37°C. The enzyme reaction was quenched by 0.1% trifluoroacetic acid.

In order to reduce the complexity of peptides, the peptides from in-solution digestion were fractionated by high pH reverse-phase high-pressure liquid chromatography (HPLC; $C_{18}$, 40 $\mu$m, 60-Å pore size; Agilent Technologies). Briefly, peptides were first separated with a gradient of 10% to 100% acetonitrile over 60 min into 60 fractions. Then, the peptides were combined into 30 fractions and dried by vacuum centrifuging.

**Liquid chromatography-MS/MS analysis.** All digested peptides were resolved in 0.1% formic acid (solvent A) and separated on an online nano-flow EASY-nLC 1200 system with a reversed-phase $C_{18}$ analytical column (15 cm by 75 $\mu$m, Thermo Fisher Scientific) at a flow rate of 300 nl/min using a linear gradient of 4 to 40% solvent B (0.1% formic acid/80% acetonitrile, vol/vol) over 90 min. The peptides were analyzed on Orbitrap Elite or Orbitrap Fusion Lumos mass spectrometers (Thermo Fisher Scientific). The full MS scan ranging from 350 to 1,800 $m/z$ (in Elite) or 350 to 1,550 (in Lumos) was acquired at 60,000 resolution at 200 $m/z$. Following every survey scan, up to 20 most intense precursor ions were selected for MS/MS fragmentation by higher-energy collision dissociation (HCD) with normalized collision energy of 32% in Elite or 35% in Lumos. For Elite acquisition, isolation width was set to 2 $m/z$ and activation time was 0.1 ms. The dynamic exclusion duration of precursor ion was set to be 120 s with a repeat count of one and $\pm$10 ppm exclusion window. For Lumos acquisition, precursor ion isolation width was set to 1.6 $m/z$ and dynamic exclusion was set to 30 s. The maximum injection times for both full MS and MS/MS were set to 50 ms and 35 ms, respectively.

The digested peptides were also separated on an Eksigent nanoLC 400 system with a 150 $\mu$m by 10 cm analytical column ($C_{18}$, 1.9 $\mu$m). Peptides were eluted at 600 nl/min flow rate with a 90-min linear solvent gradient from 8% to 35% solvent B (0.1% formic acid/80% acetonitrile, vol/vol). Peptides were then analyzed on a Triple TOF 6600 mass spectrometer (AB SCIEX) and ionized with 2,300 V ion spray voltage, declustering potential 100 V, nebulizer gas 5 lb/in$^2$, curtain gas 35 lb/in$^2$, and interface heater temperature 150°C. A full MS scan from 350 to 1,500 $m/z$ was acquired in an information-dependent acquisition (IDA) mode. The top 40 most intense precursor ions were selected for following MS/MS fragmentation. The maximum injection time for both full MS and MS/MS was 50 ms. Dynamic exclusion was set for a period of 18 s.

**Proteogenomic analysis.** All acquired MS data were submitted to the GAPP software for peptide and protein identification (34). First, the genome sequence of *Nostoc* 7120 was downloaded from the NCBI database (https://ncbi.nlm.nih.gov/genome/13531?genome_assembly_id=300961) and the protein reference database was downloaded from CyanoBase at http://genome.microbedb.jp/cyanobase/ (containing 6,135 proteins, released 2004). Then, a customizable proteogenomic database was created by translating the complete genome in six frames. The protein sequence length of more than 20 amino acids was filtered for the six-frame-translated genome database (205,978 sequences). Sequences of common laboratory contaminants, including trypsin and human keratins, were also added to the query databases. Finally, all obtained raw MS/MS data were converted to MGF format by MSConvert tool in ProteoWizard software (version 3.0.4416) (69) and then searched against the six-frame-translated genome database and protein reference database using the search engines incorporated into GAPP software (34), including Comet (70), MS-GF+ (71), and X!Tandem (72). Two additional search engines, pFind (version 3.1) (35) and MASCOT (version 2.3) (36), were also used for database-searching analysis to enhance coverage of the peptides and proteins. The parameters for database search were as follows: (i) peptide mass tolerance was set 10 ppm, (ii) fragment mass tolerance was set 0.05 Da, (iii) fixed modifications was carbamidomethyl (Cys), (iv) variable modifications were acetylation (protein N-terminal), oxidation (Met), and deamidation (Asn/Gln), and (v) trypsin/P was specified as cleavage enzyme allowing up to 2 max missing cleavages. We used the search algorithm defaults for the other parameters not common across search algorithms. The peptides identified from these search algorithms were mapped to the reference protein database and the six-frame-translated genome database, respectively, by BLASTP. The peptides matching only the six-frame-translated database were designated genome search-specific peptides (GSSPs). All peptide spectrum matches (PSMs) were filtered by a stringent false-discovery rate (FDR) threshold (FDR < 1%) in GAPP.

**PTMs identification.** To describe the protein posttranslational modifications (PTMs) landscape of *Nostoc* 7120, an unrestrictive database search algorithm, MODa (MODification via alignment) (73) was applied to find all known and even possibly unknown types of PTMs with mass shift up to 250 Da per peptide. The parameters were set as follows: (i) two maximum missed cleavage sites were allowed for trypsin, (ii) the precursor ion mass tolerance was 10 ppm and the fragment ion mass tolerance was 0.05 Da, (iii) BlindMode was 2, (iv) minModSize was −250 and maxModSize was +250, and (v) fixed posttranslational modification was carbamidomethylation (C). All other parameters were set as defaults.

Subsequently, a restrictive database search strategy, MaxQuant (version 1.6.1.0) (74), was performed to confirm the localization accuracy of the modification sites. All MS/MS raw data were used to search individually for 27 selected PTMs against the reference protein database. Enzyme specificity was set as full cleavage by trypsin/P. Fixed PTM was carbamidomethylation (Cys) apart from oxidation to nitro of Trp/Tyr (W/Y) and S-nitrosylation of Cys. Oxidation (Met), acetylation of protein N terminus, and deamidation of Asn/Gln were set as variable modifications, except that deamidation of Asn/Gln was not set as dynamic modification for monomethylation of Glu/Gln. The mass tolerance for precursor ions was set as 20 ppm in first search and 4.5 ppm in main search, and the mass tolerance for fragment ions was set as 0.02 Da. The estimated FDR threshold for PSM, proteins, and modification sites was set at 0.01. The target dynamic modifications and the enzyme max missed cleavages were set as follows: (i) phosphorylation of Ser/Thr/Tyr, two missed, (ii) monomethylation of Cys, two missed, (iii) monomethylation of Glu/Gln, two missed, (iv) monomethylation of Lys/Arg, six missed, (v) dimethylation of Lys/Arg, six missed, (vi) trimethylation of Lys, six missed, (vii) acetylation

of Lys, six missed, (viii) succinylation of Lys, six missed, (ix) butyrylation, crotonylation, malonylation, propionylation of Lys, six missed, (x) biotinylation of Lys, six missed, (xi) persulfide of Cys/Asp, two missed, (xiii) oxidation to nitro of Trp/Tyr, two missed, (xiii) S-nitrosylation of Cys, two missed, (xiv) diphthamide of His, two missed, (xv) farnesylation of Cys, two missed, (xvi) myristoylation of Cys/Lys, two missed, (xvii) palmitoylation of Ser/Thr/Cys/Lys, two missed, (xviii) ADP-ribose addition of Cys/Asp/Glu/Lys/Asn/Arg/Ser, two missed, (xix) beta-methylthiolation of Asp, two missed, (xx) hydroxymethylation of Asn, two missed, (xxi) hydroxytrimethylation of Lys, six missed, (xxii) glutarylation of Lys, six missed, (xxiii) benzoylation of Lys, six missed, (xxiv) lactylation of Lys, six missed. Finally, the modified peptides with a score of more than 40 and the modification sites with a localization probability of more than 0.75 were selected for further PTM analysis.

**RNA-seq analysis.** The transcriptome data used in this study were all retrieved from NCBI database http://ncbi.nlm.nih.gov/sra (accession numbers SRX039128, SRX039130, and SRX039131) and submitted by Flaherty et al. (49). The RNA reads were mapped to the genome of *Nostoc* 7120 using bowtie2 (75). Fragments per kilobase per million (FPKM) values were calculated by cufflinks (v2.2.1) (76), and the read count value of each gene was calculated by python script HTSeq (77). Differential expression analysis was achieved by in-house R script, and the *P* value (Fisher's exact test of ≤0.05) was used as the cutoff to define differentially expressed genes (DEGs).

**Real-time PCR analysis.** Total RNA was extracted from the cells grown under nitrogen starvation conditions for 0, 3, 12, and 24 h using TRIzol reagent (Invitrogen) according to the manufacturer's protocols. The quality and concentration of total RNA were measured by Nanodrop 2000 spectrophotometer. RT-PCR was performed using the SYBR green PCR master mix (Applied Biosystems) and the LightCycler 480 real-time PCR system (Roche). The primers of 26 novel genes were listed in Table S7B. *rnpB* was selected as the endogenous control as described previously (78). Each experiment was carried out with three independent biological replicates.

**Western blotting.** Total protein was extracted from the cells cultured under nitrogen starvation conditions for 0 and 24 h as described above. Protein concentration was measured using the BCA protein assay kit (Beyotime Biotechnology). The constant protein amounts (30 $\mu$g) from different conditions were separated by a 12% SDS-PAGE gel and transferred to polyvinylidene difluoride (PVDF) membrane (Millipore) for western blotting. The membrane was blocked in 5% bovine serum albumin BSA for 2 h at room temperature and then incubated with primary antibodies (anti-mono/di-methyllysine, anti-trimethyllysine, anti-acetyllysine, anti-propionyllysine, anti-phosphotyrosine, anti-glutaryllysine, anti-2-hydroxyisobutyryllysine, and anti-crotonyllysine) at 1:2,000 dilution overnight at 4°C. After washing three times with Tris-buffered saline with Tween 20 (TBST) buffer (25 mM Tris-HCl [pH 8.0], 150 mM NaCl, 0.1% Tween 20), the membrane was continually incubated with horseradish peroxidase-conjugated anti-rabbit IgG (1:5,000 dilutions) at room temperature for 1 h. The antibody response signal was monitored by ECL system (Advansta Inc., Menlo Park, CA).

**Novel peptides validation by using synthetic peptides.** In order to experimentally verify the credibility of these novel events, 25 GSSPs were selected and synthesized by Bioyeargene Biotechnology (Wuhan, China). Peptides were diluted using 0.1% formic acid, and approximately 1 pmol of synthetic peptide was separately subjected to LC-MS/MS analysis on LTQ-Orbitrap Elite mass spectrometer. The fragments distributions of 25 synthetic peptides were manually validated by comparing them with those from the identified peptides in proteogenomic analysis. The sequences of 25 synthetic peptides are listed in Table S7A.

**Protein preparation, tandem mass tag labeling, and LC-MS/MS analysis.** Protein lysates were extracted from the cells grown under nitrogen starvation conditions for 0, 3, and 12 h and quantified as described above. Each experiment was performed with three independent biological replicates. Equal amounts of protein extracts (100 $\mu$g) from different conditions were then used for in-solution tryptic digestion according to the protocol described in "Trypsin digestion and prefractionation" above. The tryptic peptides were subsequently labeled according to the manufacturer's protocol for the 10-plex TMT labeling kits (Thermo Fisher Scientific). Equal amounts of labeled peptides were mixed from 9 samples and fractionated into 60 fractions by high pH reverse-phase HPLC using a $C_{18}$ column (5 $\mu$m particles, 10 mm inside diameter [i.d.], 250 mm length) with a 60 min gradient of 8% to 32% acetonitrile (pH 9.0). Finally, the peptides were combined into 18 fractions and desalted with $C_{18}$ (40 $\mu$m, 60-Å particle; Agilent Technologies) columns.

After dissolving with 1% formic acid, the labeled peptides from each fraction were separated on an online nano-flow EASY-nLC 1200 system with a reversed-phase $C_{18}$ analytical column (15 cm by 75 $\mu$m, Thermo Fisher Scientific) at a flow rate of 350 nl/min using a linear gradient of 6% to 36% solvent B (0.1% formic acid/80% acetonitrile, vol/vol) over 40 min. Subsequently, the peptides were ionized by a nanospray ion source and analyzed on an Orbitrap Fusion Lumos mass spectrometer (Thermo Fisher Scientific). The full MS scan range from 350 to 1,550 *m/z* was acquired at 60,000 resolution with a minimum signal intensity of 5,000. Following each round of scanning, the top 20 most intense precursor ions were selected for MS/MS fragmentation by higher-energy collision dissociation (HCD) with normalized collision energy of 35%. Precursor ion isolation width was set to 1.6 *m/z*, and the default charge was set to 2 to 5. The dynamic exclusion duration of precursor ion was set to be 30 s with a repeat count of one and ±10 ppm exclusion window. The maximum injection times for both full MS and MS/MS were set to 50 ms and 200 ms, respectively.

**Protein identification and quantification.** Tandem MS spectra were searched against the protein database of current annotation in CyanoBase (Current CyanoBase) and an updated protein database incorporating our findings in this work (updated database), respectively, by using MaxQuant search engine (version 1.6.1.0) (74). Enzyme specificity was set as full cleavage by trypsin with two maximum missed cleavage sites permitted. Carbamidomethylation (Cys) was set as fixed modification, whereas

dynamic modifications were set as oxidation (Met), deamidation (Asn/Gln), and acetylation (protein N terminus). The mass deviation was set to 20 ppm for the first and 6 ppm for the main search. Peptide sequences smaller than six amino acids were excluded. Match-between-runs feature in MaxQuant was used for the identification with the default parameters. The false-discovery rate (FDR) thresholds for peptide and protein identifications were specified at maximum 1%, and minimum score for peptides was set to >40. Proteins identified based upon the presence of at least two unique peptides were accepted and reported. Razor and unique peptides were used for quantification. Statistical evaluation of the protein group file from MaxQuant result was performed with Perseus software (version 1.4.1.3) (79) and Microsoft Excel. The intensity of the TMT reporter ion for each protein was normalized against the total intensity of each sample. A two-sample Student's $t$ test was used for the statistical evaluation, and the DEPs (differentially expressed proteins) were defined as fold change of $\geq 1.2$ or $\leq 0.83$ and $P$ value of $<0.05$.

**Bioinformatics analysis.** Functional annotation for novel genes was performed by Blast2GO bioinformatics platform (80), and the subcellular localization was analyzed by CELLO web tool (81). The identified proteins were mapped to pathways by using Kyoto Encyclopedia of Genes and Genomes (KEGG) database (82). Protein conservative domain analysis was searched against Pfam database (83). The eggNOG-mapper was used for COG analysis (84). GO term parse, Venn diagram, sequence motif, and heatmap analysis were all performed by TBtools (85). Homology analysis was carried out by using protein BLAST online and *ClustalW* (86). The panoramic locations of all identified peptides in the genome of *Nostoc* 7120 were graphically depicted using Circos software (87). The signal peptide was predicted by SignalP 5.0 (46) and PrediSi (45) tool, and the motif was visualized by using WebLogo tool (88).

**Data availability.** All the raw MS data, the quantitative proteomic results, and the validation of GSSPs by synthesized peptides were uploaded to the public access iProX database (https://iprox.org) with the identifier IPX0002995000.

## SUPPLEMENTAL MATERIAL

Supplemental material is available online only.

**SUPPLEMENTAL FILE 1**, XLSX file, 9.4 MB.
**SUPPLEMENTAL FILE 2**, XLSX file, 0.1 MB.
**SUPPLEMENTAL FILE 3**, XLSX file, 1.9 MB.
**SUPPLEMENTAL FILE 4**, XLSX file, 1 MB.
**SUPPLEMENTAL FILE 5**, XLSX file, 0.5 MB.
**SUPPLEMENTAL FILE 6**, XLSX file, 0.3 MB.
**SUPPLEMENTAL FILE 7**, XLSX file, 0.02 MB.
**SUPPLEMENTAL FILE 8**, XLSX file, 2.4 MB.
**SUPPLEMENTAL FILE 9**, XLSX file, 0.01 MB.
**SUPPLEMENTAL FILE 10**, PDF file, 2.4 MB.

## ACKNOWLEDGMENTS

This work was supported by the National Key Research and Development Program of China (2020YFA0907400), the Chinese Academy of Sciences Grant QYZDY-SSW-SMC004, the Chinese Academy of Sciences Key Technology Talent Program (to M.Y.). We thank Ming Wang and Shuzhao Jia for their help in proteomic experiments and the Analysis and Testing Center of Institute of Hydrobiology.

F.G. and W.M. conceived and designed the experiments and wrote the article. M.Y., S.Y., and Q.Z. performed experiments and analyzed the data. J.X. and M.Y. contributed reagents/materials/analysis tools and reviewed drafts of the paper. All authors read and approved the final article.

We declare no conflicts of interest.

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
