## [Reviewer comments · Microbiology Spectrum]

Microbiology Spectrum

Proteogenomic analysis provides novel insight into genome annotation and nitrogen metabolism in *Nostoc* sp. PCC 7120

Shengchao Yu, Mingkun Yang, Jie Xiong, Qi Zhang, Xinxin Gao, Wei Miao, and Feng Ge

Corresponding Author(s): Feng Ge, Institute of Hydrobiology, Chinese Academy of Sciences

Review Timeline:

Submission Date:	June 4, 2021
Editorial Decision:	July 12, 2021
Revision Received:	August 8, 2021
Accepted:	August 16, 2021

Editor: Vincent Denef

Reviewer(s): The reviewers have opted to remain anonymous.

Transaction Report:

DOI: <https://doi.org/10.1128/Spectrum.00490-21>

July 12, 2021

Dr. Feng Ge
Institute of Hydrobiology, Chinese Academy of Sciences
Wuhan
China

Re: Spectrum00490-21 (Proteogenomic analysis provides novel insight into genome annotation and nitrogen metabolism in *Nostoc* sp. PCC 7120)

Dear Dr. Feng Ge:

Thank you for submitting your manuscript to Microbiology Spectrum. When submitting the revised version of your paper, please provide (1) point-by-point responses to the issues raised by the reviewers as file type "Response to Reviewers," not in your cover letter, and (2) a PDF file that indicates the changes from the original submission (by highlighting or underlining the changes) as file type "Marked Up Manuscript - For Review Only". Please use this link to submit your revised manuscript - we strongly recommend that you submit your paper within the next 60 days or reach out to me. Detailed information on submitting your revised paper are below.

Link Not Available

Sincerely,

Vincent Deneff

Journals Department
Reviewer comments:

Reviewer #1 (Public repository details (Required)):

There is a large dataset of Mass Spectrometry analysis of Nostoc proteins. The authors state that the data are deposited in iProX database with the identifier IPX0002995000.

However, the data are not publicly available at present. They should be available upon acceptance

Reviewer #1 (Comments for the Author):

In this work the authors use MS technology to identify proteins in Nostoc in different conditions. The high depth proteomic analysis allows the refinement of the annotation, identify new ORFs and refine the N' terminal of many proteins. In addition, they identify PTMs in a systematic way resulting in the discovery of new PTMs not previously known in bacteria. Both the new annotation and the PTM data represent an invaluable repository of information for the scientific community.

However, there are several major issues to be solved.

1- The authors used as reference the Cyanobase annotation. However, there is a newer annotation in NCBI (https://www.ncbi.nlm.nih.gov/nucore/NC_003272.1). Some of the ORFs the authors claim to have "discovered" are annotated in NCBI, as detailed below. This does not represent a methodological problem for the analysis of the MS data because in addition to the Cyanobase they use the six-frame genome database. But they should screen their GSSP against NCBI. In fact, in lines 287-289 they mention that the "novel" refined model for alr5269 was already reported in NCBI.

2-The results of this work allows for an updated annotation of ORFs in Nostoc. This is one of the most useful contributions of this work. However, the updated annotation is not easily available from the data presented. The authors should present the annotation as a table similar to table S1E with two additional columns: the updated coordinates for each ORF, and the Refseq new name of each ORF, if available.

3-There is an overall confusion between newly identified proteins and confirmation of proteins already annotated.

4-The "new" genes in Fig 5B, C and updated 5' end in 5E are already annotated in RefSeq.

5- There is disagreement in Fig. 6 between transcriptomic and RT-PCR. An example out of several: RNAseq: NG-2 increases at 12 h; RT-PCR: NG-2 decrease at 12 h.

6-Furthermore, there is disagreement between RNA data in Fig. 6 and protein quantification data in Fig.7. For instance, NG-7 protein amount increase upon nitrogen deprivation, but both RNASeq and RT-PCR indicate that there is a reduction in the amount of mRNA. The authors should at least discuss these discrepancies.

Other points

Line 41: photosynthesis

Lines 171-173: When describing the shared proteins identified they should mention in the text the three heterocyst specific proteins with split genes that are in Table S2.

Line 182-185: I find this speculation baseless.

Lines 199-203. The statements in this paragraph are not relevant because the authors do not perform a functional enrichment analysis of the different PTMs. Table 4B reflects just the general distribution of proteins.

Line 216-217. The PTM data are clearly very interesting, novel, and are one of the main strengths of the paper. From figure 4C it seems that at least propionyl and crotonyl modifications are altered in the -N sample for some proteins. This is a very interesting result that should be highlighted in the text.

Line 267: Insert here reference 50 so that the reader knows what RNASeq data are used.

Line 273. ¿Is "transcriptomic evidence" here the same than "RNA-Seq evidence" in 267?

Line 336 NG-1, NG2-and NG-7 are already present in NCBI RefSeq. Only NG-6 is novel.
Lines 361-363, As indicated above in relation to lines 182-185, this speculation is baseless. I don't understand the meaning of "weaker protein coding potential".

Line 45. References 42 and 33 are wrong.

Line 46. Rather than reference 66, the following reference could be more appropriate: Elhai, J., and Khudyakov, I. (2018). Ancient association of cyanobacterial multicellularity with the regulator HetR and an RGSGR pentapeptide-containing protein (PatX). *Mol Microbiol.* doi:10.1111/mmi.14003 or Khudyakov, I., Gladkov, G., and Elhai, J. (2020). Inactivation of Three RG(S/T)GR Pentapeptide-Containing Negative Regulators of HetR Results in Lethal Differentiation of *Anabaena* PCC 7120. *Life* 10. doi:10.3390/life10120326.

Line 410. NG-20 is not novel. In fact, the authors refer to a 2018 paper [67] about this protein. Several of the "novel" ORFs in Table S6 are already annotated in Refseq.

Paragraph starting at line 412. The extended discussion on proteins related to heavy metal metabolism (including fig. 8) is rather out of the main focus of the paper, could be deleted or strongly reduced.

Table S1E has a recurrent typo in column D: putative "proteion".

References should be carefully reviewed. References 10 and 41 are duplicated. References 66 and 101 are incomplete or wrongly formatted.

Reviewer #2:

In addition to the detailed comments above, another expert in the field raised concerns regarding the use of this strain, claiming it is a mutant and highly mutable. Please comment on the time this strain has been in culture since the initial genome sequence and consider this in your interpretation of differences between the initially deposited sequence annotation and the update you provide.

Staff Comments:

Preparing Revision Guidelines

For complete guidelines on revision requirements, please see the Instructions to Authors at [link to

page]. **Submissions of a paper that does not conform to Microbiology Spectrum guidelines will delay acceptance of your manuscript.**

Please return the manuscript within 60 days; if you cannot complete the modification within this time period, please contact me. If you do not wish to modify the manuscript and prefer to submit it to another journal, please notify me of your decision immediately so that the manuscript may be formally withdrawn from consideration by Microbiology Spectrum.

If you would like to submit an image for consideration as the Featured Image for an issue, please contact Spectrum staff.

RESPONSE TO REVIEWERS

Manuscript ID: Spectrum00490-21

Manuscript Title: Proteogenomic analysis provides novel insight into genome annotation and nitrogen metabolism in *Nostoc* sp. PCC 7120

Professor Vincent Denef

Editor

Microbiology Spectrum

Dear Prof. Denef:

We greatly appreciate your decision to allow us to submit the revised manuscript. We sincerely thank the editor and reviewers for the careful assessment of our manuscript and their thoughtful and constructive comments. Based on the reviewers' comments, we have revised the manuscript. In what follows, we present a point-by-point answer to the reviewers' comments. Our responses are marked with "**Author reply**".

Reviewer #1 (Public repository details (Required)):

There is a large dataset of Mass Spectrometry analysis of Nostoc proteins. The authors state that the data are deposited in iProX database with the identifier IPX0002995000. However, the data are not publicly available at present. They should be available upon acceptance.

Authors reply: We appreciate the valuable comments from the reviewer. We have submitted the data to iProX database with the identifier IPX0002995000 and they are publicly available now.

Reviewer #1 (Comments for the Author):

In this work the authors use MS technology to identify proteins in *Nostoc* in different conditions. The high depth proteomic analysis allows the refinement of the annotation, identify new ORFs and refine the N' terminal of many proteins. In addition, they

identify PTMs in a systematic way resulting in the discovery of new PTMs not previously known in bacteria. Both the new annotation and the PTM data represent an invaluable repository of information for the scientific community.

However, there are several major issues to be solved.

*1- The authors used as reference the Cyanobase annotation. However, there is a newer annotation in NCBI (https://www.ncbi.nlm.nih.gov/nucore/NC_003272.1). Some of the ORFs the authors claim to have "discovered" are annotated in NCBI, as detailed below. This does not represent a methodological problem for the analysis of the MS data because in addition to the Cyanobase they use the six-frame genome database. But they should screen their GSSP against NCBI. In fact, in lines 287-289 they mention that the "novel" refined model for *alr5269* was already reported in NCBI.*

Authors reply: We appreciate the comments of the reviewer. At present, there are three genome assemblies for *Nostoc* sp. PCC 7120 published in NCBI database. Their RefSeq assembly accessions are GCF_000009705.1 (released in 2004, annotated 6132 proteins), GCF_003990585.1 (released in 2019, annotated 5900 proteins) and GCF_014696735.1 (released in 2020, annotated 5732 proteins), respectively. Cyanobase annotation was completed by *Takakazu* with reference to GCF_000009705.1 (1), which has been widely used as a reference database in the functional genomic studies of *Nostoc* sp. PCC 7120. According to the comments of the reviewer, we performed BLAST analysis of these novelties discovered in this study with the 2019 and 2020 version of genomic annotations (accession: GCF_003990585.1 and GCF_014696735.1) collected in the NCBI database, respectively. The sequence alignment results have refreshed in the **Table S6D** and **S6E**. Of which, four novel genes (NG-3, -7, -15, -20) and twenty-three revised genes shared identical sequence with genes annotated in the latest genome of *Nostoc* sp. PCC 7120 (**Table S6D & S6E**). For the revised gene model of *alr5269*, it has been annotated in the genome assembly of 2020 version (100% sequence similarity with MBD2272597.1). While compared with the gene model annotated in the 2019 version, it has 2 amino acids extension at the N-terminus. (**Figure 5D**).

2-The results of this work allow for an updated annotation of ORFs in Nostoc. This is one of the most useful contributions of this work. However, the updated annotation is not easily available from the data presented. The authors should present the annotation as a table similar to table S1E with two additional columns: the updated coordinates for each ORF, and the Refseq new name of each ORF, if available.

Authors reply: We thank the reviewer for this constructive comment. The updated coordinates for each ORF have been displayed in the column H-J of **Table S6B** and **Table S6C**. And the Refseq new names of each ORF have been updated in **Table S6D** and **S6E**.

3-There is an overall confusion between newly identified proteins and confirmation of proteins already annotated.

Authors reply: We apologize for this lack of clarity. In this study, all raw MS data were analyzed against two databases: (I) Cyanobase database, (II) a six-frame translated genome database of *Nostoc* sp. PCC 7120. The identified peptides exclusively matching the six-frame genome database were designated as genome search-specific peptides (GSSPs). And the GSSPs were mapped to the unique genomic locus to identify open reading frames (ORFs) using BLAST. ORFs that were only mapped to the genome regions without overlap with any known gene models were designated as novel protein-coding genes, that is the newly identified proteins (NG-1~NG-26 are the newly identified proteins in this study). ORFs that partially overlapped with the annotated protein-coding regions were designated as revised gene models (NG-27~NG-101 are the revised gene models in this study). While the identified unique peptides matching with the six-frame genome database and Cyanobase database were used as confirmation of proteins already annotated (The 5519 proteins identified in the proteomic landscape of *Nostoc* 7120 are the confirmation of proteins already annotated in Cyanobase). We have marked in the in the revised manuscript (**Line 125 and Line 260**) in response to the comments of the reviewer.

4-The "new" genes in Fig 5B, C and updated 5' end in 5D are already annotated in RefSeq.

Authors reply: Thanks for the comments. As mentioned above, we have performed sequence alignment with the genome annotation of 2019 version and 2020 version and the results have showed in the **Table S6D** and **S6E**. As showed in the table, NG-15 does exist in both the 2019 and 2020 versions of genome annotations (**Figure 5B**). While NG-20 is only annotated in the 2020 versions of genome assembly (**Figure 5C**). For the revised gene model of alr5269, it has been annotated in the genome assembly of 2020 version (100% sequence similarity with MBD2272597.1). When compared with the gene model annotated in the 2019 version, it has 2 amino acids extension at the N-terminus (**Figure 5D**). As the suggestion of the reviewer, we have revised the **Figure 5** and its legend. We add 2019 annotation and 2020 annotation information in the revised **Figure 5** and explain in the legend that the accession with 2019 annotation and 2020 annotation represent that the sequence existed in the *Nostoc* 7120 genome annotation of 2019 version or 2020 version in NCBI database, respectively.

5- There is disagreement in Fig. 6 between transcriptomic and RT-PCR. An example out of several: RNAseq: NG-2 increases at 12 h; RT-PCR: NG-2 decrease at 12 h.

Authors reply: We appreciate the comment of the reviewer. We have also noticed the disagreement between transcriptomic and RT-PCR. We believe that the reasons for this divergence may be due to the different samples, different detection and analysis methods, shorter transcripts, and lower expression levels, etc (2, 3). Therefore, **Figure 6** was mainly used to show these new genes can be detected at the mRNA level and differentially expressed under nitrogen deficiency. We have mentioned this disagreement in the revised manuscript in response to the comments of the reviewers. (**Line 317-318**)

6-Furthermore, there is disagreement between RNA data in Fig. 6 and protein quantification data in Fig.7. For instance, NG-7 protein amount increase upon

nitrogen deprivation, but both RNASeq and RT-PCR indicate that there is a reduction in the amount of mRNA. The authors should at least discuss these discrepancies.

Authors reply: We thank the reviewer for this constructive suggestion. It is widely accepted mRNA level usually has a poor correlation with the corresponding protein level (4, 5). Perhaps substantial regulatory processes, including post-transcriptional, translational and degradation regulation, contribute at least as much as transcription itself in the determination of protein concentrations. We have discussed it as the suggestion of the reviewers in the revised manuscript. (**Line 349-352**)

Other points:

Line 41: photosynthesis

Authors reply: We are sorry for the mistake. We have revised the spelling, and the whole manuscript has been carefully checked.

Lines 171-173: When describing the shared proteins identified they should mention in the text the three heterocyst specific proteins with split genes that are in Table S2.

Authors reply: We appreciate the reviewer's comment. We have added it in the revised manuscript (**Line 175-177**) as the reviewer's suggestion.

Line 182-185: I find this speculation baseless.

Authors reply: We apologize for this lack of clarity. We have deleted the controversial speculation in the revised manuscript.

Lines 199-203. The statements in this paragraph are not relevant because the authors do not perform a functional enrichment analysis of the different PTMs. Table 4B reflects just the general distribution of proteins.

Authors reply: We have performed a functional enrichment analysis of the different PTMs in response to the reviewer's comment. We have provided the result as an alternative to **Figure 4B**. The relevant statements have also been revised in manuscript (**line 202-205**).

Line 216-217. The PTM data are clearly very interesting, novel, and are one of the main strengths of the paper. From figure 4C it seems that at least propionyl and crotonyl modifications are altered in the -N sample for some proteins. This is a very interesting result that should be highlighted in the text.

Authors reply: We thank the reviewer for the useful suggestion. As advised, we highlighted the result in the revised manuscript. (**Line 218-221**)

Line 267: Insert here reference 50 so that the reader knows what RNASeq data are used.

Authors reply: We thank the reviewer for the valuable suggestion. The transcriptome data used in this study were all derived from reference 50. We have added this specification to the RNA-seq analysis section of the materials and methods. (**Line 588-590**)

Line 273. Is "transcriptomic evidence" here the same than "RNA-Seq evidence" in 267?

Authors reply: We apologize for this lack of clarity. The transcriptome data used in this study were all retrieved from NCBI database <http://www.ncbi.nlm.nih.gov/sra> (accession: SRX039128, SRX039130 and SRX039131) submitted by Flaherty et al [50]. We have added this specification to the RNA-seq analysis section of the materials and methods. (**Line 588-590**)

Line 336 NG-1, NG-2 and NG-7 are already present in NCBI RefSeq. Only NG-6 is novel.

Authors reply: We appreciate the reviewer's comment. As mentioned above, there are discrepancies between different versions of genome annotation. We have showed the sequence alignment of novel genes in **Table S6D**. As showed in the table, only NG-7 is consistent with the genome annotation of 2019 and 2020 version. Although NG-1 also has homologous sequences, its N-terminus is 10 amino acids shorter than

that in the NCBI RefSeq. While neither NG-2 nor NG-6 has been annotated in the NCBI RefSeq by referring to the BLAST analysis.

Lines 361-363, As indicated above in relation to lines 182-185, this speculation is baseless. I don't understand the meaning of "weaker protein coding potential".

Authors reply: We apologize for the lack of clarity. In the present *Nostoc* 7120 genome annotation, a large number (3,525 of 6,135) of the previously predicted protein-coding genes are annotated as hypothetical protein or unknown protein. These proteins only showed similarity to hypothetical proteins, or even no significant similarity to any known proteins (1). The expression of these predicted protein-coding genes remains to be verified. While in our proteomic identification, 438 out of 616 unidentified proteins were annotated as unknown protein or hypothetical protein. Of which, only 232 putative proteins (about 54%) have functional features according to Gene Ontology (GO), COG, KEGG pathway and Pfam annotations (**Table S2C**). We speculated that these unidentified predicted encoding genes may not exist. We have deleted the controversial speculation in the revised manuscript as the reviewer's suggestion.

Line 405. References 42 and 33 are wrong.

Authors reply: We apologize for the lack of clarity. 42 and 33 here refer to the number of putative proteins, not the reference number. We have revised it in the manuscript to avoid misunderstanding. (**Line 419**)

Line 406. Rather than reference 66, the following reference could be more appropriate: Elhai, J., and Khudyakov, I. (2018). Ancient association of cyanobacterial multicellularity with the regulator HetR and an RGSGR pentapeptide-containing protein (PatX). Mol Microbiol. doi:10.1111/mmi.14003 or Khudyakov, I., Gladkov, G., and Elhai, J. (2020). Inactivation of Three RG(S/T)GR Pentapeptide-Containing Negative Regulators of HetR Results in Lethal Differentiation of Anabaena PCC 7120. Life 10. doi:10.3390/life10120326.

Authors reply: We thank the reviewer for the comment and have substituted the references according to reviewer's suggestion.

Line 410. NG-20 is not novel. In fact, the authors refer to a 2018 paper [67] about this protein. Several of the "novel" ORFs in Table S6 are already annotated in Refseq.

Authors reply: We appreciate the reviewer's comment. There are three different versions of genome annotations for *Nostoc* sp. PCC 7120 published in NCBI database as mentioned above. In this study, we select the Cyanobase annotation because it has been widely used as a reference database. As showed in **Table S6D** and **Figure 5C**, we also acknowledge that NG-20 is present in the latest genome annotation. While this protein is missing in the Cyanobase database and 2019 version of genome annotation. The sequence alignment analysis of other novelties with genes annotated in different versions of genome are also summarized in **Table S6D** and **S6E**.

Paragraph starting at line 412. The extended discussion on proteins related to heavy metal metabolism (including fig. 8) is rather out of the main focus of the paper, could be deleted or strongly reduced.

Authors reply: We thank the reviewer for this constructive suggestion. We have strongly reduced this discussion section in the revised manuscript.

Table S1E has a recurrent typo in column D: putative "proteion".

Authors reply: Thanks for pointing out these mistakes and we have corrected them in the revised manuscript.

References should be carefully reviewed. References 10 and 41 are duplicated. References 66 and 101 are incomplete or wrongly formatted.

Authors reply: We are sorry for the mistakes. We have corrected them, and the whole revised manuscript has been carefully checked.

Reviewer #2:

In addition to the detailed comments above, another expert in the field raised concerns regarding the use of this strain, claiming it is a mutant and highly mutable. Please comment on the time this strain has been in culture since the initial genome sequence and consider this in your interpretation of differences between the initially deposited sequence annotation and the update you provide.

Authors reply: We highly appreciate the comment raised by the reviewer. *Nostoc* sp. PCC 7120 is commonly used as a model strain for studying cell differentiation and multicellular pattern. At present, the widely used substrains mainly from either University of Chicago, Michigan State University, or the PCC. And the three substrains were reported to have undergone microevolution by comparative genome analysis, including single nucleotide polymorphisms (SNPs), small insertion/deletions (indels; 1 to 3 bp), fragment deletions, and transpositions [4] (**Table R1-2**). The genomic locations of novelties identified in this study were listed in **Table R3**. By comparing the regions of the novelties with the sequence mutations reported by *Xu et al*, no overlap region was found. In addition, genome microevolution events have also been reported in other organisms, such as *Escherichia coli*, *Bacillus subtilis*, *Synechocystis* sp. PCC 6803, *etc* (6-8). The genome reannotations based on proteogenomics analysis, similar to this study, have also been performed in these organisms (9-11).

The substrain used in this study was obtained in 2017 from Jindong Zhao (Peking University), who had brought it from the University of Chicago. The Cyanobase database annotation also refers to the genome sequencing of the strain from University of Chicago. Therefore, although the strain has undergone microevolution during the cultivation, these novelties identified in this study should not be caused by these mutations in view of the comparing analysis above and the methodology employed in this study is also reasonable.

Table R1. SNPs in *Nostoc* sp. PCC 7120 substrains IHB and HAU

Plasmid	Position	Reference base	IHB	depth	HAU	depth	Gene (s)	Region
---------	----------	-------------------	-----	-------	-----	-------	----------	--------

Alpha	3289	C	T	139	C	\	all7005, asr7006	Intergenic
Alpha	8242	T	T	\	C	125	all7011	ORF
Alpha	25546	T	T	\	G	159	all7027	ORF
Alpha	49297	A	A	\	G	353	alr7063	ORF
Alpha	72155	T	T	\	C	310	all7084	ORF
Alpha	77566	T	G	1,822	T	\	alr7089	ORF
Alpha	85981	A	G	2,144	G	225	all7098	ORF
Alpha	92804	G	A/G	1,554/438	G	\	all7106, alr7107	Intergenic
Alpha	125193	T	T	\	C	132	alr7129	ORF
Alpha	147600	G	G	\	A	141	asr7143	ORF
Alpha	157737	G	A	2,081	G	\	alr7157	ORF
Alpha	161185	C	T	1,205	C	\	all7160, all7161	Intergenic
Alpha	183584	C	T	2,016	T	140	all7185	ORF
Alpha	197224	T	T	\	C	188	all7191	ORF
Alpha	216050	G	C	1,931	G	\	alr7206	ORF
Alpha	216051	T	C	1,914	T	\	alr7206	ORF
Alpha	216052	T	C	1,895	T	\	alr7206	ORF
Alpha	231547	A	G	2,049	G	155	all7218, alr7219	Intergenic
Alpha	268895	T	T	\	C	323	alr7249	ORF
Alpha	289163	A	G	1,827	G	139	all7275	ORF
Alpha	300552	A	G	1,909	G	227	alr7294, alr7295	Intergenic
Alpha	300902	A	G	1,879	G	199	alr7295	ORF
Alpha	310851	G	G	\	A	175	alr7299	ORF
Alpha	326058	T	T	\	C	210	alr7304	ORF
Alpha	364530	G	G	\	A	224	asr7330	ORF
Alpha	407466	G	A	1,983	G	\	asr7385, alr7386	Intergenic
Beta	51863	G	A	483	G	\	alr7555	ORF
Gamma	23623	T	C	1,400	C	310	all8023	ORF
Gamma	56266	C	T	2,065	C	\	asl8049, asl8050	Intergenic

Gamma	56708	C	T	2,062	C	\	asl8049, asl8050	Intergenic
Gamma	96179	A	A	\	G	439	all8083, asl8084	Intergenic
Delta	32318	A	G	446	G	493	alr8542, alr8543	Intergenic
Zeta	4713	A	C	1,101	C	364	alr9504, alr9505	Intergenic
Chromosome	99793	C	C	\	T	439	alr0094	ORF
Chromosome	136013	T	T	\	C	383	alr0132, all0131	Intergenic
Chromosome	141747	A	A	\	G	414	asl0137, all0138	Intergenic
Chromosome	165432	A	A	\	G	427	all0160	ORF
Chromosome	172755	T	T	\	C	295	all0167	ORF
Chromosome	176465	G	A	414	G	\	all0168	ORF
Chromosome	228394	T	C	398	C	375	all0211	ORF
Chromosome	240374	A	G	439	G	449	alr0223	ORF
Chromosome	254220	A	G	309	A	\	alr0236, alr0237	Intergenic
Chromosome	335391	T	T	\	C	383	alr0295, all0293	Intergenic
Chromosome	367421	A	G	498	A	\	all0323	ORF
Chromosome	388027	C	T	474	C	\	alr0336	ORF
Chromosome	411083	A	G	491	G	473	all0355	ORF
Chromosome	461600	A	A	\	G	430	all0394	ORF
Chromosome	635525	A	A	\	G	467	alr0543, all0542	Intergenic
Chromosome	663464	A	G	522	G	345	alr0568	ORF
Chromosome	676298	G	G	\	T	337	asr0581, alr0582	Intergenic
Chromosome	703818	C	C	\	T	393	all0606	ORF
Chromosome	791750	T	C	478	T	\	all0684	ORF
Chromosome	804565	A	G	504	A	\	asr0697	ORF
Chromosome	838235	G	G	\	A	371	alr0719	ORF
Chromosome	850685	T	T	\	C	421	all0729	ORF
Chromosome	873358	T	C	568	T	\	alr0751, alr0752	Intergenic
Chromosome	964001	G	G	\	A	427	alr0838	ORF
Chromosome	965275	T	C	461	C	454	alr0840, all0839	Intergenic

Chromosome	1105390	C	T	524	C	\	alr0950, alr0951	Intergenic
Chromosome	1164680	C	C	\	T	299	all0993	ORF
Chromosome	1181589	T	C	441	C	389	all1011	ORF
Chromosome	1197843	T	C	477	C	404	all1027	ORF
Chromosome	1213151	T	T	\	C	43	alr1042, alr1041	Intergenic
Chromosome	1218823	C	T/C	355/89	C	\	all1047	ORF
Chromosome	1233544	G	G	\	A	462	all1058	ORF
Chromosome	1234860	A	G	472	A	\	all1059	ORF
Chromosome	1260414	A	G	480	G	416	all1076	ORF
Chromosome	1276768	A	G	535	G	474	all1089	ORF
Chromosome	1291183	A	G	409	G	421	all1101	ORF
Chromosome	1297308	C	C	\	T	439	alr1108	ORF
Chromosome	1297479	A	G	431	G	373	alr1108	ORF
Chromosome	1313571	T	T	\	C	416	alr1121	ORF
Chromosome	1326135	C	T	468	C	\	alr1128	ORF
Chromosome	1384940	G	G	\	A	414	all1177	ORF
Chromosome	1425785	T	T	\	C	418	all1210	ORF
Chromosome	1449194	A	G	462	G	432	all1227	ORF
Chromosome	1467026	T	C	548	C	446	alr1236	ORF
Chromosome	1469539	T	T	\	C	491	all1237, alr1238	Intergenic
Chromosome	1473304	A	G	449	A	\	alr1240	ORF
Chromosome	1503377	A	G	532	G	389	alr1266	ORF
Chromosome	1511804	G	G	\	A	425	all1272	ORF
Chromosome	1524963	G	A	497	A	407	all1281	ORF
Chromosome	1526058	T	C	479	C	436	alr1282	ORF
Chromosome	1589551	G	G	\	A	392	all1338	ORF
Chromosome	1603350	G	A	378	A	429	alr1348	ORF
Chromosome	1611061	A	G	492	A	\	all1357	ORF
Chromosome	1611129	A	G	480	A	\	all1357	ORF

Chromosome	1689791	A	G/A	446/93	A	\	all1427	ORF
Chromosome	1720294	G	G	\	A	379	all1463, alr1462	Intergenic
Chromosome	1740294	G	A	405	G	\	all1477	ORF
Chromosome	1740843	T	C	455	C	401	all1478	ORF
Chromosome	1771480	A	A	\	C	412	all1509	ORF
Chromosome	1810252	T	T	\	C	443	all1549	ORF
Chromosome	1816020	T	C	433	T	\	all1553	ORF
Chromosome	1829742	G	T	515	G	\	alr1564	ORF
Chromosome	1839174	C	T	447	T	386	all1574, alr1575	Intergenic
Chromosome	1895931	G	A	468	A	386	alr1614	ORF
Chromosome	1897575	G	A	500	A	379	alr1614	ORF
Chromosome	1907745	A	A	\	G	370	alr1619	ORF
Chromosome	1908542	T	C	460	T	\	alr1620	ORF
Chromosome	1939174	C	T	488	C	\	all1639	ORF
Chromosome	1968465	T	T	\	C	342	all1649	ORF
Chromosome	1981387	G	A	473	A	464	alr1659	ORF
Chromosome	1992410	A	A	\	G	389	alr1669	ORF
Chromosome	2008739	G	A	480	A	432	all1683	ORF
Chromosome	2020785	G	A	528	A	499	all1691, all1692	Intergenic
Chromosome	2027656	G	A	500	G	\	all1695	ORF
Chromosome	2028469	C	T	486	C	\	all1695	ORF
Chromosome	2095482	A	G	474	A	\	alr1742	ORF
Chromosome	2103674	A	A	\	G	446	asl1749	ORF
Chromosome	2199111	T	T	\	C	455	alr1833, alr1834	Intergenic
Chromosome	2203380	C	T	488	T	456	asl1839	ORF
Chromosome	2212421	C	C	\	T	389	all1849	ORF
Chromosome	2232920	C	T	465	C	\	alr1870	ORF
Chromosome	2310473	A	G	240	A	\	alr1926, alr1927	Intergenic
Chromosome	2348508	A	T	433	A	\	alr1965, alr1966	Intergenic

Chromosome	2359208	T	C	420	C	410	all1974	ORF
Chromosome	2386615	G	G	\	A	421	all1990	ORF
Chromosome	2466963	C	T	458	T	360	all2058	ORF
Chromosome	2479725	T	G	460	G	440	alr2073, all2072	Intergenic
Chromosome	2498543	C	C	\	A	244	alr2090	ORF
Chromosome	2498548	T	T	\	C	248	alr2090	ORF
Chromosome	2556356	T	T	\	C	463	alr2130	ORF
Chromosome	2571912	A	A	\	G	429	alr2143	ORF
Chromosome	2580643	G	G	\	A	385	all2149	ORF
Chromosome	2665829	T	C	523	C	407	all2221, all2222	Intergenic
Chromosome	2677570	G	A	534	G	\	alr2233	ORF
Chromosome	2756402	A	G	462	G	355	all2287	ORF
Chromosome	2832449	G	G	\	A	401	alr2350, all2349	Intergenic
Chromosome	2834963	A	A	\	G	435	all2352	ORF
Chromosome	2843120	C	T	465	C	\	alr2361	ORF
Chromosome	2856383	T	C	462	C	358	alr2373	ORF
Chromosome	2871033	C	C	\	T	430	all2384	ORF
Chromosome	2902538	G	G	\	A	431	alr2418	ORF
Chromosome	2927969	T	T	\	C	395	alr2434	ORF
Chromosome	2966029	C	C	\	T	428	alr2467	ORF
Chromosome	2979695	A	A	\	G	436	alr2481	ORF
Chromosome	3067783	A	G/A	370/93	A	\	all2567	ORF
Chromosome	3150434	G	A	477	A	439	all2635	ORF
Chromosome	3164466	T	C	414	C	426	all2643	ORF
Chromosome	3214501	A	G	545	G	464	all2655	ORF
Chromosome	3244445	A	G	406	G	424	all2675, all2676	Intergenic
Chromosome	3280388	A	G	447	G	404	all2688	ORF
Chromosome	3280989	A	A	\	G	422	all2689	ORF
Chromosome	3292733	A	A	\	G	408	all2699	ORF

Chromosome	3299861	A	G	436	G	366	all2706	ORF
Chromosome	3314107	C	T	494	T	434	alr2719, alr2718	Intergenic
Chromosome	3321812	C	T	448	T	407	alr2725	ORF
Chromosome	3376560	T	C	510	C	455	asl2779, alr2780	Intergenic
Chromosome	3384352	A	G	485	G	411	alr2784, alr2785	Intergenic
Chromosome	3385448	C	T	491	T	374	alr2785	ORF
Chromosome	3388341	T	A	329	A	356	all2787	ORF
Chromosome	3404883	C	T	469	T	435	alr2800	ORF
Chromosome	3439866	T	C	479	C	445	alr2824	ORF
Chromosome	3449008	C	C	\	T	403	alr2832	ORF
Chromosome	3461181	G	G	\	A	433	alr2840	ORF
Chromosome	3461281	T	C/T	413/103	T	\	alr2840	ORF
Chromosome	3522353	C	T	487	T	314	alr2884, all2883	Intergenic
Chromosome	3529532	A	G	444	G	337	all2891	ORF
Chromosome	3553554	T	T	\	C	392	all2911	ORF
Chromosome	3561805	T	T	\	C	437	alr2920, alr2921	Intergenic
Chromosome	3665879	T	C	439	C	337	asl3025	ORF
Chromosome	3678170	A	G	444	A	\	alr3037	ORF
Chromosome	3678824	A	G	466	G	436	alr3037	ORF
Chromosome	3682278	C	T	407	T	439	all3040, all3041	Intergenic
Chromosome	3707449	A	G	424	A	\	alr3059	ORF
Chromosome	3717999	T	T	\	C	407	alr3068	ORF
Chromosome	3902869	T	C	476	T	\	all3232	ORF
Chromosome	3997485	T	C/T	387/50	T	\	alr3311	ORF
Chromosome	4004306	C	T	434	C	\	all3314	ORF
Chromosome	4054871	T	T	\	C	442	alr3351	ORF
Chromosome	4070477	G	G	\	A	340	alr3363	ORF
Chromosome	4071943	T	T	\	C	397	alr3364	ORF
Chromosome	4074557	C	T	455	T	320	alr3366, all3367	Intergenic

Chromosome	4085038	C	T	449	T	463	all3375	ORF
Chromosome	4096661	T	C	469	C	375	alr3385	ORF
Chromosome	4105348	A	A	\	G	366	alr3397	ORF
Chromosome	4206630	T	T	\	C	416	alr3491	ORF
Chromosome	4211631	G	A	490	A	365	alr3497	ORF
Chromosome	4219140	A	G	486	A	\	all3503	ORF
Chromosome	4283858	A	G	453	A	\	alr3553, alr3554	Intergenic
Chromosome	4329437	G	A	479	A	431	alr3584	ORF
Chromosome	4372460	A	C	71	C	472	alr3620	ORF
Chromosome	4383161	C	T/C	517/59	C	\	all3632	ORF
Chromosome	4429266	A	G	294	A	\	alr3672, all3673	Intergenic
Chromosome	4524010	G	G	\	A	455	all3746	ORF
Chromosome	4542759	A	A	\	G	388	alr3761	ORF
Chromosome	4580258	C	T	452	C	\	alr3789	ORF
Chromosome	4591842	G	A	500	G	\	alr3799, alr3800	Intergenic
Chromosome	4609347	G	A	409	G	\	alr3811, alr3812	Intergenic
Chromosome	4624534	A	A	\	G	395	alr3825	ORF
Chromosome	4633553	G	A	460	A	417	alr3832, alr3833	Intergenic
Chromosome	4658414	G	A	515	G	\	all3859	ORF
Chromosome	4667926	A	A	\	G	460	alr3867	ORF
Chromosome	4692046	G	A	455	G	\	all3891	ORF
Chromosome	4707809	A	G	522	A	\	all3903, alr3904	Intergenic
Chromosome	4708391	A	A	\	G	426	alr3904	ORF
Chromosome	4741099	G	A	484	A	387	all3927	ORF
Chromosome	4834014	G	A/G	437/131	G	\	asl4014	ORF
Chromosome	4835685	C	T	490	C	\	alr4016	ORF
Chromosome	4853972	T	C	480	C	418	alr4028	ORF
Chromosome	4862032	C	T	488	T	430	all4035	ORF
Chromosome	4930827	T	T	\	C	470	all4092	ORF

Chromosome	4942838	C	C	\	T	396	all4102	ORF
Chromosome	4960654	T	T	\	C	418	all4117	ORF
Chromosome	4987803	T	C	453	T	\	alr4141, all4142	Intergenic
Chromosome	5012714	C	T	526	T	473	alr4166	ORF
Chromosome	5031775	C	C	\	T	389	all4182	ORF
Chromosome	5036789	T	C	478	C	420	all4188	ORF
Chromosome	5053532	T	T	\	C	432	alr4216	ORF
Chromosome	5054846	C	A	488	C	\	all4218	ORF
Chromosome	5069810	T	C	417	C	434	all4233	ORF
Chromosome	5072713	A	G	513	G	361	all4236	ORF
Chromosome	5093978	T	T	\	C	397	alr4247	ORF
Chromosome	5096718	A	G	452	G	378	all4248	ORF
Chromosome	5118388	T	C	494	C	486	alr4268	ORF
Chromosome	5123906	T	C	487	T	\	alr4273	ORF
Chromosome	5128378	T	C	473	C	467	alr4275	ORF
Chromosome	5130579	T	T	\	C	355	alr4277	ORF
Chromosome	5148689	A	G/A	376/85	A	\	all4294	ORF
Chromosome	5220452	C	T	484	C	\	all4358	ORF
Chromosome	5268442	G	A	461	G	\	alr4394	ORF
Chromosome	5277885	C	T	511	C	\	all4402	ORF
Chromosome	5295867	G	T	489	G	\	alr4417	ORF
Chromosome	5299799	G	A	541	A	397	asr4421	ORF
Chromosome	5366378	A	C	466	A	\	all4480	ORF
Chromosome	5376257	T	C	532	T	\	alr4489	ORF
Chromosome	5451412	A	A	\	G	457	all4556	ORF
Chromosome	5472869	A	G	174	A	\	all4578	ORF
Chromosome	5472870	A	C	174	A	\	all4578	ORF
Chromosome	5501951	A	G	458	G	450	alr4604	ORF
Chromosome	5542778	A	G	462	G	423	all4639	ORF

Chromosome	5596861	C	T	482	T	462	all4690	ORF
Chromosome	5636415	T	T	\	C	344	all4729	ORF
Chromosome	5638397	A	G	465	G	391	all4731	ORF
Chromosome	5646409	A	G	464	G	437	alr4734, all4735	Intergenic
Chromosome	5656553	A	G	493	G	461	asl4743	ORF
Chromosome	5677447	G	G	\	T	469	all4763	ORF
Chromosome	5718687	A	G	427	G	387	all4799	ORF
Chromosome	5730046	C	T	487	T	442	alr4811, alr4812	Intergenic
Chromosome	5746803	C	C	\	T	420	all4828	ORF
Chromosome	5808428	A	G	470	G	437	alr4877	ORF
Chromosome	5873706	C	A	526	C	\	all4925	ORF
Chromosome	5895688	C	T	484	C	\	alr4938, alr4939	Intergenic
Chromosome	5930524	C	C	\	T	399	all4968	ORF
Chromosome	6040371	T	T	\	C	353	alr5068	ORF
Chromosome	6053276	G	G	\	A	398	all5082, alr5081	Intergenic
Chromosome	6053727	T	T	\	C	381	all5082, alr5081	Intergenic
Chromosome	6077149	C	T	509	C	\	all5100, alr5101	Intergenic
Chromosome	6078691	T	T	\	C	406	alr5101	ORF
Chromosome	6085980	G	A	491	G	\	all5105	ORF
Chromosome	6087384	G	T	443	G	\	all5106	ORF
Chromosome	6103380	G	G	\	A	435	all5113	ORF
Chromosome	6150516	C	T	478	C	\	all5153	ORF
Chromosome	6239409	G	A	531	G	\	alr5225	ORF
Chromosome	6263798	T	C	438	C	423	alr5249	ORF
Chromosome	6275081	G	G	\	A	377	alr5259	ORF
Chromosome	6276832	G	A	527	A	367	asr5261	ORF
Chromosome	6352567	C	T/C	441/61	C	\	all5323	ORF
Chromosome	6365449	G	A/G	426/92	G	\	alr5331	ORF
Chromosome	6375751	C	T	465	C	\	all5342	ORF

Chromosome	6386500	T	T	\	C	448	alr5351	ORF
Chromosome	6387506	A	G	482	G	481	alr5351	ORF

Note: HAU, substrain from Huazhong Agriculture University; IHB, substrain from Institute of Hydrobiology.

Table R2. Transpositions and fragment deletions in genomes of substrains

substrain	HAU		IHB		HAU/IHB	HAU/IHB/CPW
mutation	fragment deletions	lacked insertion sequences	fragment deletions	lacked insertion sequences	acquired ISs	short sequence stretches
chromosome position	2030740 to 2031696	1286797 to 1288267	4616790 to 4618012	1286797 to 1288267	3579748	1551763-1551764; 2243236-2243237 ; 4247463-4247464
alpha plasmid position	861 to 1715; 175281 to 176938; 312401 to 313185	242000 to 243358	97521 to 99371; 160056 to 161094	242000 to 243358	45585/504 56; 165178	\
delta plasmid position	\	6937 to 8620	\	6937 to 8620	\	\
gamma plasmid position	\	\	\	\	518; 25703	\

Note: HAU, substrain from Huazhong Agriculture University; IHB, substrain from Institute of Hydrobiology; CPW, substrain from C. Peter Wolk.

Table R3. List of the locations of novelties in the genome of *Nostoc* sp. PCC 7120

Category	Name	Start	Stop	Strand	chromosome/plasmid	overlap with the mutant region
Intergenic	NG-1	3061381	3061743	+	Chromosome	NO
	NG-2	50856	50945	-	Chromosome	NO
	NG-3	2175325	2175567	+	Chromosome	NO
	NG-4	4743923	4744180	-	Chromosome	NO
	NG-5	291906	291977	-	Chromosome	NO
	NG-6	3326403	3326579	+	Chromosome	NO
	NG-7	6329546	6329734	+	Chromosome	NO
	NG-8	1979	2275	+	zeta	NO
	NG-9	28264	28458	+	delta	NO
Different_frame	NG-10	6174084	6174314	+	Chromosome	NO
	NG-11	1916410	1916619	-	Chromosome	NO
	NG-12	1480942	1481019	+	Chromosome	NO
	NG-13	3304145	3304204	+	Chromosome	NO
	NG-14	2363297	2363404	+	Chromosome	NO
	NG-15	41129	41383	+	delta	NO
Opposite_strand	NG-16	1848810	1849055	-	Chromosome	NO
	NG-17	3549321	3549479	+	Chromosome	NO
	NG-18	1606753	1606842	-	Chromosome	NO
	NG-19	5523306	5523383	+	Chromosome	NO
	NG-20	3937924	3938082	-	Chromosome	NO
	NG-21	552087	552458	+	Chromosome	NO
	NG-22	3593954	3594163	+	Chromosome	NO
	NG-23	3960188	3960316	-	Chromosome	NO
	NG-24	1935534	1935599	+	Chromosome	NO
	NG-25	3430865	3430978	-	Chromosome	NO
	NG-26	955647	955844	+	Chromosome	NO

N-terminal	NG-27	5602199	5602834	-	Chromosome	NO
	revised	NG-28	5563575	5564075	-	Chromosome
	NG-29	5476611	5478074	-	Chromosome	NO
	NG-30	6398394	6400049	+	Chromosome	NO
	NG-31	2398714	2399418	-	Chromosome	NO
	NG-32	557581	560325	-	Chromosome	NO
	NG-33	4387092	4388429	-	Chromosome	NO
	NG-34	9135	9635	-	Chromosome	NO
	NG-35	2806617	2807138	-	Chromosome	NO
	NG-36	4437976	4439016	-	Chromosome	NO
	NG-37	2720541	2721353	+	Chromosome	NO
	NG-38	2845637	2846836	-	Chromosome	NO
	NG-39	1927877	1929331	+	Chromosome	NO
	NG-40	4890360	4892165	+	Chromosome	NO
	NG-41	4449359	4450171	+	Chromosome	NO
	NG-42	183303	184679	-	Chromosome	NO
	NG-43	6285585	6287060	+	Chromosome	NO
	NG-44	161063	161677	-	Chromosome	NO
	NG-45	3040425	3041672	+	Chromosome	NO
	NG-46	3458379	3459263	+	Chromosome	NO
	NG-47	1892178	1892783	-	Chromosome	NO
	NG-48	1062152	1063162	-	Chromosome	NO
	NG-49	5141638	5142366	-	Chromosome	NO
	NG-50	4174473	4176371	+	Chromosome	NO
	NG-51	4113746	4114192	+	Chromosome	NO
	NG-52	439610	440356	-	Chromosome	NO
	NG-53	542438	543202	-	Chromosome	NO
	NG-54	1091458	1092084	-	Chromosome	NO
	NG-55	5566391	5567062	-	Chromosome	NO

NG-56	752322	753362	-	Chromosome	NO
NG-57	186587	188383	-	Chromosome	NO
NG-58	5605481	5606170	+	Chromosome	NO
NG-59	5317808	5318593	-	Chromosome	NO
NG-60	1361762	1362160	+	Chromosome	NO
NG-61	3795929	3796582	+	Chromosome	NO
NG-62	5073517	5074065	-	Chromosome	NO
NG-63	4949471	4950196	-	Chromosome	NO
NG-64	5174268	5175236	+	Chromosome	NO
NG-65	6062857	6064005	-	Chromosome	NO
NG-66	397206	398111	+	Chromosome	NO
NG-67	5108554	5109243	+	Chromosome	NO
NG-68	4660695	4662368	-	Chromosome	NO
NG-69	6134827	6136002	-	Chromosome	NO
NG-70	903541	905127	-	Chromosome	NO
NG-71	5046358	5046708	-	Chromosome	NO
NG-72	567494	568564	+	Chromosome	NO
NG-73	2090923	2091618	+	Chromosome	NO
NG-74	3936567	3937862	+	Chromosome	NO
NG-75	3754820	3755926	+	Chromosome	NO
NG-76	2851805	2852293	-	Chromosome	NO
NG-77	3834074	3834370	+	Chromosome	NO
NG-78	3121525	3122052	+	Chromosome	NO
NG-79	1335319	1335696	+	Chromosome	NO
NG-80	2161853	2162545	-	Chromosome	NO
NG-81	940332	940649	-	Chromosome	NO
NG-82	6403133	6403798	-	Chromosome	NO
NG-83	6054759	6057398	+	Chromosome	NO
NG-84	5105966	5106448	-	Chromosome	NO

NG-85	2454539	2455384	+	Chromosome	NO
NG-86	3232857	3233654	+	Chromosome	NO
NG-87	2639519	2640103	-	Chromosome	NO
NG-88	3580698	3581294	+	Chromosome	NO
NG-89	51286	53541	-	Chromosome	NO
NG-90	5499142	5499570	+	Chromosome	NO
NG-91	2251178	2252299	+	Chromosome	NO
NG-92	4854080	4855237	+	Chromosome	NO
NG-93	4967988	4969247	+	Chromosome	NO
NG-94	5746427	5746699	-	Chromosome	NO
NG-95	3341288	3342556	+	Chromosome	NO
NG-96	623032	623289	+	Chromosome	NO
NG-97	5788304	5791381	-	Chromosome	NO
NG-98	290858	291880	-	Chromosome	NO
NG-99	5699561	5700901	+	Chromosome	NO
NG-100	57260	57430	-	gamma	NO
NG-101	84271	85470	+	alpha	NO

Reference

1. Kaneko T, Nakamura Y, Wolk CP, Kuritz T, Sasamoto S, Watanabe A, Iriguchi M, Ishikawa A, Kawashima K, Kimura T, Kishida Y, Kohara M, Matsumoto M, Matsuno A, Muraki A, Nakazaki N, Shimpo S, Sugimoto M, Takazawa M, Yamada M, Yasuda M, Tabata S. 2001. Complete genomic sequence of the filamentous nitrogen-fixing cyanobacterium *Anabaena* sp. strain PCC 7120. *DNA Res* 8:205-13; 227-53.
2. Everaert C, Luypaert M, Maag JLV, Cheng QX, Dinger ME, Hellemans J, Mestdagh P. 2017. Benchmarking of RNA-sequencing analysis workflows using whole-transcriptome RT-qPCR expression data. *Sci Rep* 7:1559.
3. Robert C, Watson M. 2015. Errors in RNA-Seq quantification affect genes of relevance to human disease. *Genome Biol* 16:177.
4. de Sousa Abreu R, Penalva LO, Marcotte EM, Vogel C. 2009. Global signatures of protein and mRNA expression levels. *Mol Biosyst* 5:1512-26.
5. Vogel C, Marcotte EM. 2012. Insights into the regulation of protein abundance from proteomic and transcriptomic analyses. *Nat Rev Genet* 13:227-32.
6. Barrick JE, Yu DS, Yoon SH, Jeong H, Oh TK, Schneider D, Lenski RE, Kim JF. 2009.

Genome evolution and adaptation in a long-term experiment with *Escherichia coli*. *Nature* 461:1243-7.

7. Shi L, Derouiche A, Pandit S, Rahimi S, Kalantari A, Futo M, Ravikumar V, Jers C, Mokkaapati V, Vlahoviček K, Mijakovic I. 2020. Evolutionary Analysis of the *Bacillus subtilis* Genome Reveals New Genes Involved in Sporulation. *Mol Biol Evol* 37:1667-1678.
8. Kanesaki Y, Shiwa Y, Tajima N, Suzuki M, Watanabe S, Sato N, Ikeuchi M, Yoshikawa H. 2012. Identification of substrain-specific mutations by massively parallel whole-genome resequencing of *Synechocystis* sp. PCC 6803. *DNA Res* 19:67-79.
9. Krug K, Carpy A, Behrends G, Matic K, Soares NC, Macek B. 2013. Deep coverage of the *Escherichia coli* proteome enables the assessment of false discovery rates in simple proteogenomic experiments. *Mol Cell Proteomics* 12:3420-30.
10. Pettersen VK, Steinsland H, Wiker HG. 2015. Improving genome annotation of enterotoxigenic *Escherichia coli* TW10598 by a label-free quantitative MS/MS approach. *Proteomics* 15:3826-34.
11. Ravikumar V, Nalpas NC, Anselm V, Krug K, Lenuzzi M, Šestak MS, Domazet-Lošo T, Mijakovic I, Macek B. 2018. In-depth analysis of *Bacillus subtilis* proteome identifies new ORFs and traces the evolutionary history of modified proteins. *Sci Rep* 8:17246.

August 16, 2021

Dr. Feng Ge
Institute of Hydrobiology, Chinese Academy of Sciences
Wuhan
China

Re: Spectrum00490-21R1 (Proteogenomic analysis provides novel insight into genome annotation and nitrogen metabolism in *Nostoc* sp. PCC 7120)

Dear Dr. Feng Ge:

Thank you for your detailed modifications to the manuscript. Please note that while you have provided accession numbers to data, you should provide the following in a "Data Availability" paragraph at the end of the Materials and Methods section of full-length articles (or at the end of the text in shorter article types): data description, name of the repository, and DOIs or accession numbers.

Your manuscript has been accepted, and I am forwarding it to the ASM Journals Department for publication. You will be notified when your proofs are ready to be viewed.

Sincerely,

Vincent Deneff
Editor, Microbiology Spectrum

Journals Department
Table S6: Accept

Table S4: Accept

Table S1: Accept

Table S8: Accept

Table S9: Accept

Table S5: Accept

Table S2: Accept

Table S3: Accept

Figure S1-S6: Accept

Table S7: Accept